# Projecting the Response of Greenland's Peripheral Glaciers to Future Climate Change: Glacier Losses, Sea Level Impact, Freshwater Contributions, and Peak Water Timing

Muhammad Shafeeque[1,2,3,*], Jan-Hendrik Malles[1,4], Anouk Vlug[1,5], Marco Möller[1,2,6], Ben Marzeion[1,2,*]

[1]Climate Lab, Institute of Geography, University of Bremen, 28359 Bremen Germany
[2]MARUM – Center for Marine Environmental Sciences, University of Bremen, 28359 Bremen Germany
[3]Alfred Wegener Institute for Polar and Marine Research, 27570 Bremerhaven, Germany
[4]Institute of Environmental Physics, University of Bremen, 28359 Bremen Germany
[5]Department of Atmospheric and Cryospheric Sciences, University of Innsbruck, 6020 Innsbruck, Austria
[6]Geodesy and Glaciology, Bavarian Academy of Sciences and Humanities, 80539 Munich, Germany

*Correspondence to*: Muhammad Shafeeque (shafeequ@uni-bremen.de) and Ben Marzeion (ben.marzeion@uni-bremen.de)

**Abstract.** Greenland's peripheral glaciers are significant contributors to sea level rise and freshwater fluxes, yet their future evolution remains poorly constrained. This study projects the response of these glaciers to future climate change using the Open Global Glacier Model (OGGM) forced by CMIP6 climate data under four emission scenarios. By 2100, the glaciers are projected to lose 19 ± 6 % (SSP126) to 44 ± 15 % (SSP585) of their area and 29 ± 6 % (SSP126) to 52 ± 14 % (SSP585) of their volume (ensemble mean ± 1 standard deviation across 10 GCMs), contributing 10 ± 2 to 19 ± 5 mm to sea level rise. Solid ice discharge is projected to decrease, while freshwater runoff will peak within the 21st century. The runoff composition is projected to change drastically, with shares of glacier ablation decreasing from 92 % in 2021-2030 to 72 % by 2091-2100 and shares of rainfall and snowmelt increasing 8-fold and 15-fold, respectively, suggesting a shift in the hydrological regime. Timing of the maximum runoff varies across scenarios (2050 ± 21 for SSP126; 2082 ± 9 for SSP585) and subregions, with the projected maximum runoff reaching 214-293 Gt/yr, implying significantly increased future freshwater fluxes. These changes will impact fjord water characteristics and coastal hydrography, and potentially influence larger ocean circulation patterns.

**Keywords:** Climate Change; Greenland's Peripheral Glaciers; Freshwater; Ice Discharge; Sea Level Rise; OGGM; Peak Water

## 1 Introduction

The Arctic region has experienced a significant increase in air temperatures in recent decades, warming nearly four times faster than the global average (Rantanen et al., 2022). This rapid warming profoundly impacts Greenland's peripheral glaciers, which are either completely detached from the ice sheet or dynamically decoupled (Rastner et al., 2012). These glaciers exhibit accelerated responses to warming compared to the slower-responding ice sheet (Khan et al., 2022; Noel et al., 2017; Larsen et al., 2022; Bolch et al., 2013; Larocca et al., 2023), which are linked to increased surface ablation and solid ice discharge, indicating a high sensitivity to atmospheric warming and oceanic forcing (Bjørk et al., 2017; Liu et al., 2022a; Möller et al.,

2024). Greenland's peripheral glaciers account for 9-11 % of the global glacier volume outside Antarctica and the Greenland Ice Sheet, and they are significant contributors to current and future sea level rise, presently delivering the second largest

contribution (10-13 %) to sea level rise originating from the global glaciers outside the two ice sheets (Hugonnet et al., 2021; Bolch et al., 2013). The peripheral glaciers are equivalent to only approximately 5 % of the area and less than 1 % of the volume of the Greenland Ice Sheet, yet they contribute 11-20 % of Greenland's total ice mass loss (Hugonnet et al., 2021; Khan et al., 2022; Bollen et al., 2023; Bolch et al., 2013).

Despite their significance, the evolution of these glaciers under future climate scenarios remains insufficiently explored,

particularly with respect to a partitioning of freshwater contributions to sea level rise, i.e., solid ice discharge and freshwater runoff. This distinction is critical for predicting changes in fjord water characteristics, sea level, and oceanic circulation (Hopwood et al., 2020; Sugiyama et al., 2021; Edwards et al., 2021; Mankoff et al., 2020; Nowicki et al., 2020). Both solid ice discharge and freshwater runoff (surface melting and rainfall) directly contribute to sea level rise when they enter the ocean (Edwards et al., 2021; Hopwood et al., 2020). However, they differ in timing and spatial distribution of their contributions.

When marine-terminating glaciers (excluding floating tongues) calve icebergs into the fjords, these icebergs immediately contribute to sea level rise. As the icebergs drift away from the glacier and gradually melt, they release freshwater over a larger area and longer time scale (Bamber et al., 2018; Davison et al., 2020; Enderlin et al., 2021). Liquid freshwater also directly contributes to rising sea levels when the water enters the ocean. This freshwater input is more concentrated near the glacier terminus and has a more immediate effect on sediment transport, fjord characteristics, and local sea level (Beckmann et al.,

2019; Slater et al., 2020). Understanding the dynamics and interplay of solid ice discharge and surface liquid freshwater from peripheral glaciers is crucial for accurately assessing Greenland's overall ice mass losses and their impacts under future climate change.

Existing studies often overlook the impact of future climate change on the individual components of freshwater contributions from these peripheral glaciers and how these changes in magnitude and timing propagate to affect fjord water characteristics,

ocean circulation, and sea level rise (Cowton et al., 2015; Hopwood et al., 2020). Solid ice discharge from peripheral glaciers represents approximately 2.6% to 5.3% of total mass loss from marine-terminating Greenland peripheral glaciers when considering observed dynamic mass loss as a component of total mass loss from Greenland's terrestrial ice (Bollen et al., 2023; Malles et al., 2023). Projections accounting for frontal ablation processes can result in an 8% increase in marine-terminating glacier mass loss compared to projections without frontal ablation (Malles et al., 2023), yet this process has received less

attention when modeling future climate change scenarios. The composition of future liquid freshwater fluxes from Greenland's periphery, including the relative contributions of ice melt, snowmelt and rainfall, remains poorly quantified (Mernild et al., 2010; Mernild et al., 2013; Mernild et al., 2018). The changes in magnitude and timing of freshwater composition in the surrounding ocean impact the ocean circulation and marine ecosystems (Perner et al., 2019; Bamber et al., 2018; Hopwood et al., 2020; Mankoff et al., 2020; Mathis and Mikolajewicz, 2020; Kanzow et al., 2024). Moreover, the timing of the maximum

runoff (called peak water from here on), which has major implications for ocean circulation patterns, fjord ecosystems, and sea level, also requires dedicated projections of freshwater fluxes and timing focused on the peripheral glaciers rather than the

whole ice sheet (Oliver et al., 2018; Aschwanden et al., 2019; Bliss et al., 2014). This distinction is important because peripheral glaciers and the Greenland Ice Sheet are likely to exhibit different peak water timing. While the massive ice sheet may continue to increase its meltwater contribution well beyond this century, smaller and more climate-sensitive peripheral glaciers are expected to reach peak water earlier. Consequently, some fjords primarily fed by peripheral glaciers may experience peak water within the projection period of this study, while others dominated by ice sheet runoff may not. Previous research suggests that certain glaciers may have already transitioned towards a more cold-based regime (Carrivick et al., 2023). Despite a warming regional climate, this transition occurs because glacier thinning reduces driving stress and ice velocities, making pressure melting at the glacier bed less likely while allowing winter cold to penetrate more easily to the bed, causing the glacier to freeze to its substrate and move more slowly. This shift implies a potential change in the timing of meltwater release. By focusing on peripheral glaciers, we can better understand and anticipate localized changes in freshwater input to coastal areas, which is crucial for assessing impacts on fjord ecosystems, coastal dynamics, and potentially larger ocean circulation patterns.

This study aims to address these research gaps by investigating how Greenland's peripheral glaciers will evolve under different future climate change scenarios, considering spatial and temporal variability. It employs the Open Global Glacier Model (OGGM) (Maussion et al., 2019), calibrated with recent geodetic mass balance data covering 2000-2020 (Hugonnet et al., 2021) and satellite-derived observational frontal ablation data covering the same period (Kochtitzky et al., 2022), and is forced using an ensemble of ten GCMs from CMIP6 (Eyring et al., 2016) under four emission scenarios (SSP126, SSP245, SSP370, SSP585) ranging from low to high emissions, with projections extending from 2020 to 2100. Our modeling results yield projections of future mass loss of Greenland's peripheral glaciers, including the ability to distinguish between mass loss occurring above and below sea level. This distinction allows for more accurate estimations of their contributions to sea level rise, as well as detailed projections of both solid and liquid freshwater contributions. Furthermore, we project the timing and magnitude of peak runoff for these glaciers. Thus, our study also gives insights into the changing composition of projected liquid freshwater runoff, including the relative contributions of different sources such as ice melt, snowmelt, and rainfall, which contributes to enhance our understanding of the evolving hydrological dynamics and their implications in the region.

## 2 Materials and Methods

### 2.1 Study Region: Greenland's Peripheral Glaciers

This study focuses on Greenland's peripheral glaciers that have been classified into three different connectivity levels (CL) by Rastner et al. (2012): completely detached from the ice sheet (CL0), dynamically decoupled (CL1), and dynamically connected to the ice sheet (CL2). In our study, we only consider glaciers of categories CL0 and CL1 (Fig. 1a), as glaciers in category CL2 are usually considered to be part of the ice sheet (Hock et al., 2019; Marzeion et al., 2020). Glacier outlines are taken from the Randolph Glacier Inventory (RGI) version 6.0 (Pfeffer et al., 2014). Deviating from this inventory, we adopted an enhanced subdivision comprising individual drainage basins for the Flade Isblink Ice Cap (FIIC; RGI ID: RGI60-05.10315)

in Northeast Greenland. The new subdivision of FIIC (Fig. 1b) encompasses several marine-terminating basins; however, based on velocity observations, only six of them are active calving basins (Recinos et al., 2021; Möller et al., 2022). Active calving basins are those with measurable ice velocities at the terminus and evidence of ongoing calving activity based on satellite observations, while inactive basins have negligible terminus velocities and show no recent calving activity, based on velocity data and findings from Recinos et al. (2021) and Möller et al. (2022). This study groups the peripheral glaciers into seven regions: North-East, Central-East, South-East, South-West, Central-West, North-West, and North (Fig. 1).

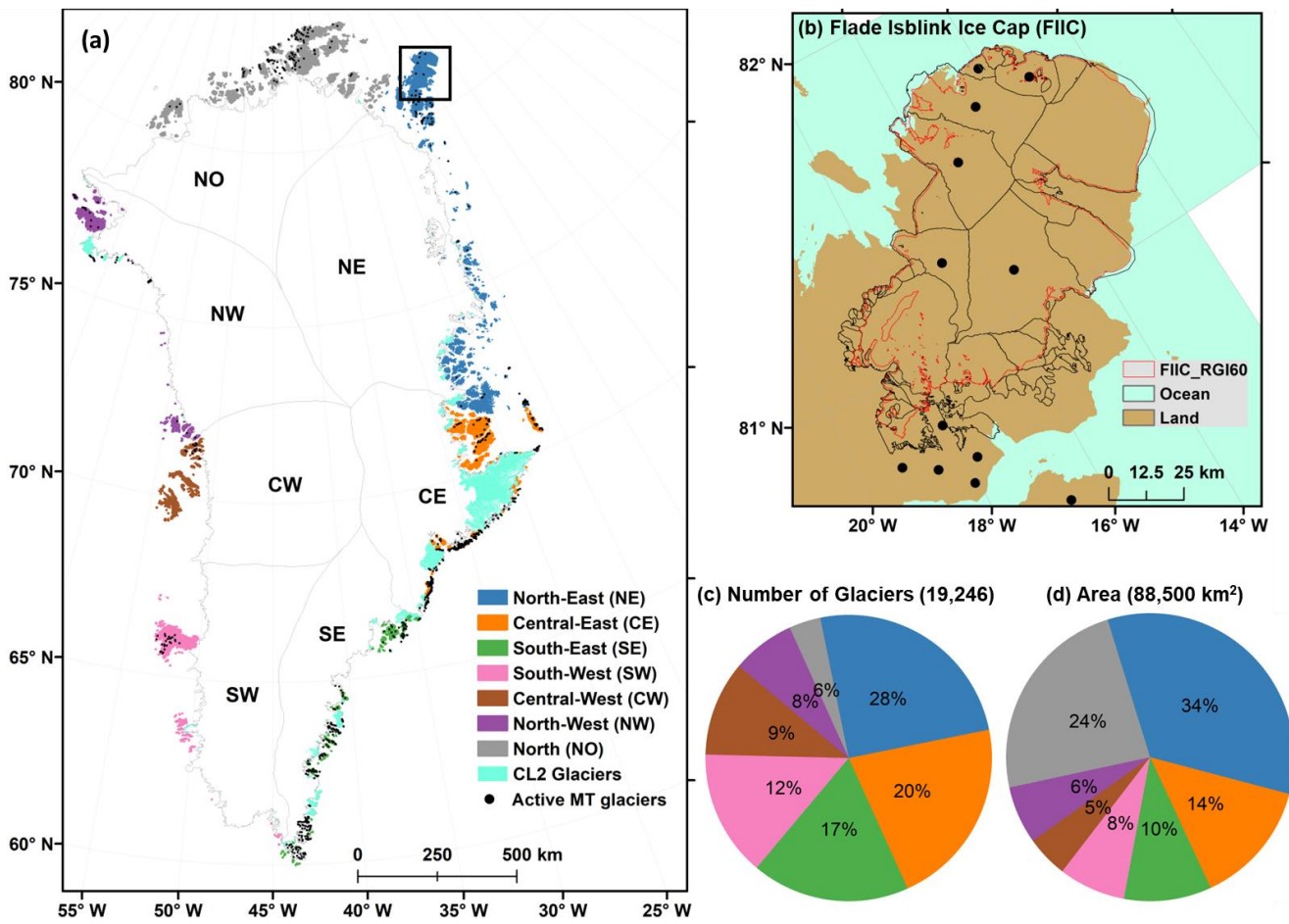

**Figure 1. Spatial distribution and characteristics of Greenland's peripheral glaciers. (a) Distribution of the considered peripheral glaciers (connectivity levels: CL0 and CL1) across different subregions, excluding CL2 glaciers. Active marine-terminating (MT) glaciers are shown as black dots. (b) New subdivision of the FIIC showing active marine-terminating glaciers; the red outline represents the FIIC boundary as a single entity in RGI6.0. (c) Total number of glaciers and their percentage in each subregion. (d) Total glacier area and its percentage across the subregions of Greenland. The order of subregions in the pie charts follow their approximate geographic position around Greenland periphery.**

## 2.2 Data

### 2.2.1 Climate Data (ERA5, CMIP6) and Preprocessing

ERA5 climate data (monthly air temperature and precipitation) (Hersbach et al., 2020) were used as boundary conditions to
calibrate the mass balance model. A multiplicative precipitation correction factor (with no vertical gradientr; see details in
Section 2.3.2) was applied within the OGGM mass balance module. This correction accounts for processes like orographic
precipitation, avalanches, and wind-blown snow, which are not resolved by the ERA5 data (Maussion et al., 2019).
CMIP6 data for ten GCMs (Table 1) and four Shared Socioeconomic Pathways (SSPs: SSP126, SSP245, SSP370, and SSP585)
are used to force the model from 2020 until 2100. Among the selected SSPs, SSP126 represents sustainability (low emissions),
SSP245 middle-of-the-road development, SSP370 regional rivalry, and SSP585 fossil-fueled growth (high emissions), each
offering distinct scenarios for future global socioeconomic development and associated climate challenges (Riahi et al., 2017).
The selected GCMs have been employed in several previous studies for similar glacier projections (Edwards et al., 2021;
Malles et al., 2023; Rounce et al., 2023; Zekollari et al., 2024), chosen based on their performance in simulating key climatic
variables relevant to glacier dynamics and their ability to represent a broad range of potential future climates (Walsh et al.,
2018). This standardized selection of GCMs provides consistency and continuity, facilitates comparison with the growing
body of global glacier literature, and enables participation in coordinated model intercomparison projects like GlacierMIP
(Hock et al., 2019; Marzeion et al., 2020). The ensemble approach using ten carefully selected GCMs provides a statistically
robust representation of climate uncertainty while maintaining comparability with earlier studies. Furthermore, the sample size
is large enough to encompass a wide range of potential climatic futures, thus yielding a robust set of scenarios and increasing
confidence in the projections. Although CMIP6 models generally do not include dynamic ice sheet components (Eyring et al.,
2016; Nowicki et al., 2016), our glacier model OGGM explicitly accounts for ice dynamics. The climate data from these GCMs
serves as input for OGGM, rather than directly modeling glacier evolution.

**Table 1. Selected GCMs from CMIP6 for future climate change data until 2100.**

| GCM | Variant | Spatial resolution (°) | Temporal coverage | Reference |
|---|---|---|---|---|
| BCC-CSM2-MR | r1i1p1f1 | 1.12 | 1850-2100 | Xin et al. (2018) |
| CAMS-CSM1-0 | r1i1p1f1 | 1.12 | 1850-2100 | Rong (2019) |
| FGOALS-f3-L | r1i1p1f1 | 1.00 | 1850-2100 | Yu (2019) |
| CESM2-WACCM | r1i1p1f1 | 1.25 | 1850-2100 | Danabasoglu (2019) |
| GFDL-ESM4 | r1i1p1f1 | 1.25 | 1850-2100 | Horowitz et al. (2018) |
| INM-CM4-8 | r1i1p1f1 | 2.00 | 1850-2100 | Volodin et al. (2019a) |
| INM-CM5-0 | r1i1p1f1 | 2.00 | 1850-2100 | Volodin et al. (2019b) |
| MPI-ESM1-2-HR | r1i1p1f1 | 0.94 | 1850-2100 | Von Storch et al. (2017) |
| MRI-ESM2-0 | r1i1p1f1 | 1.12 | 1850-2100 | Yukimoto et al. (2019) |
| NorESM2-MM | r1i1p1f1 | 1.25 | 1850-2100 | Seland et al. (2020) |

The CMIP6 2-m temperature and total precipitation data are downscaled to the baseline climate ERA5, that has been used for
calibrating OGGM. A variation of the delta method (e.g., Ramírez Villegas and Jarvis, 2010) is being used for this procedure,

whereby the precipitation is scaled and scaled temperature anomalies are applied to the 1981-2020 baseline climatology. The delta method applies scaled temperature anomalies and scaled precipitation ratios on a month-by-month basis to the baseline climatology. The scaling ensures that the variability (standard deviation) of the bias-corrected temperature and precipitation

matches that of the ERA5. For temperature:

$$T_{corrected} = T_{ERA5} + scf \times \left( T_{GCM} - \overline{T_{GCM\,(1981-2020)}} \right) \tag{1}$$

where the scaling factor is:

$$scf = \frac{std\left(T_{ERA5\,(1981-2020)}\right)}{std\left(T_{GCM\,(1981-2020)}\right)} \tag{2}$$

For precipitation:

$$P_{corrected} = P_{ERA5} \times \left( \frac{P_{GCM}}{\overline{P_{GCM\,(1981-2020)}}} \right) \tag{3}$$

This bias correction methodology effectively removes systematic GCM biases while preserving the climate change signal, ensuring that local Greenland climate variability is accurately represented. OGGM applies this bias-corrected climate data to glacier-specific elevation profiles using a constant temperature lapse rate of -6.5 K/km (OGGM default; Maussion et al., 2019), within the range commonly used for Arctic glacier applications (Gardner et al., 2009).

**2.2.2 Glacier, Elevation, Mass Balance and Frontal Ablation Data**

OGGM requires information about the location, area, terminus type, and elevation of each glacier at some point in time (usually the date of data acquisition) within the modeled time interval. These data were taken from RGI 6.0 (Pfeffer et al., 2014). For topographic data, we used the ArcticDEM dataset (Porter et al., 2018) for most of our study's glaciers and the GIMP DEM (Howat et al., 2014) to fill in the gaps.

This study utilizes the mass change estimates for each glacier in the RGI 6.0 during 2000-2020, provided by Hugonnet et al. (2021). However, these mass changes are based on differences in surface elevations derived from digital elevation models (DEMs) between different points in time and do not include any changes occurring below sea level. Thus, when estimating total mass changes and calibrating models of marine-terminating glaciers, it is essential to correct for the mass budget disregarded by not considering changes below sea level.

To obtain frontal ablation estimates, including the mass changes below sea level, which are needed to prevent an erroneous calibration of the surface mass balance model in OGGM, we use the satellite-derived dataset from Kochtitzky et al. (2022). These frontal ablation estimates are used to correct the mass budget for marine-terminating glaciers, ensuring accurate calibration of the surface mass balance model. For a detailed description of how this data is incorporated into the calibration process, see Section 2.4.

## 2.3 Open Global Glacier Model (OGGM)

### 2.3.1 Model Framework and Setup

OGGM is a numerical model framework designed to simulate the evolution of glaciers on a basin to global scale. It is based on a combination of physical and empirical equations that relate glacier mass balance, ice flow, and geometry to environmental variables, such as temperature, precipitation, and topography (Maussion et al., 2019). This study uses OGGM v1.5.3 (Maussion et al., 2022) with custom implementations for frontal ablation based on Malles et al. (2023). The basic flowchart of OGGM setup, calibration, and run as used in this study is presented in Fig. 2.

The topographic data is interpolated and resampled to a resolution suitable for the glacier size, then smoothed using a Gaussian filter, and finally reprojected centered on the individual glacier using Transverse Mercator map projection. OGGM automatically determines grid resolution based on glacier area using $\Delta x = 14\sqrt{S} + 10$, where $\Delta x$ is the grid spatial resolution (m) and S is glacier area (km²). The grid resolution is bounded by 10 m (minimum) and 200 m (maximum), ensuring smaller glaciers receive higher resolution for better geometric representation while larger glaciers use coarser resolution for computational efficiency.

OGGM uses a flowline model based on Shallow Ice Approximation (SIA) to simulate the ice dynamics (Maussion et al., 2019). This flowline considers the width of the glacier, allowing the model to match the observed area-elevation distribution of real glaciers and to parametrize changes in glacier width with thickness changes. This study uses the binned elevation-band flowlines method (Werder et al., 2019). The mean of the slopes within a quantile range is used to calculate the glacier's slope, removing outliers and accurately representing the glacier's main tongue and true length. The downstream lines and bed shape are also calculated to allow the glacier to grow. The dynamical simulations commence from the date of the glacier's data acquisition in the RGI. The starting date of the simulations may thus vary over a few years between glaciers. The initial geometry comprises the surface area specified by the RGI and the outcome of the ice thickness inversion.

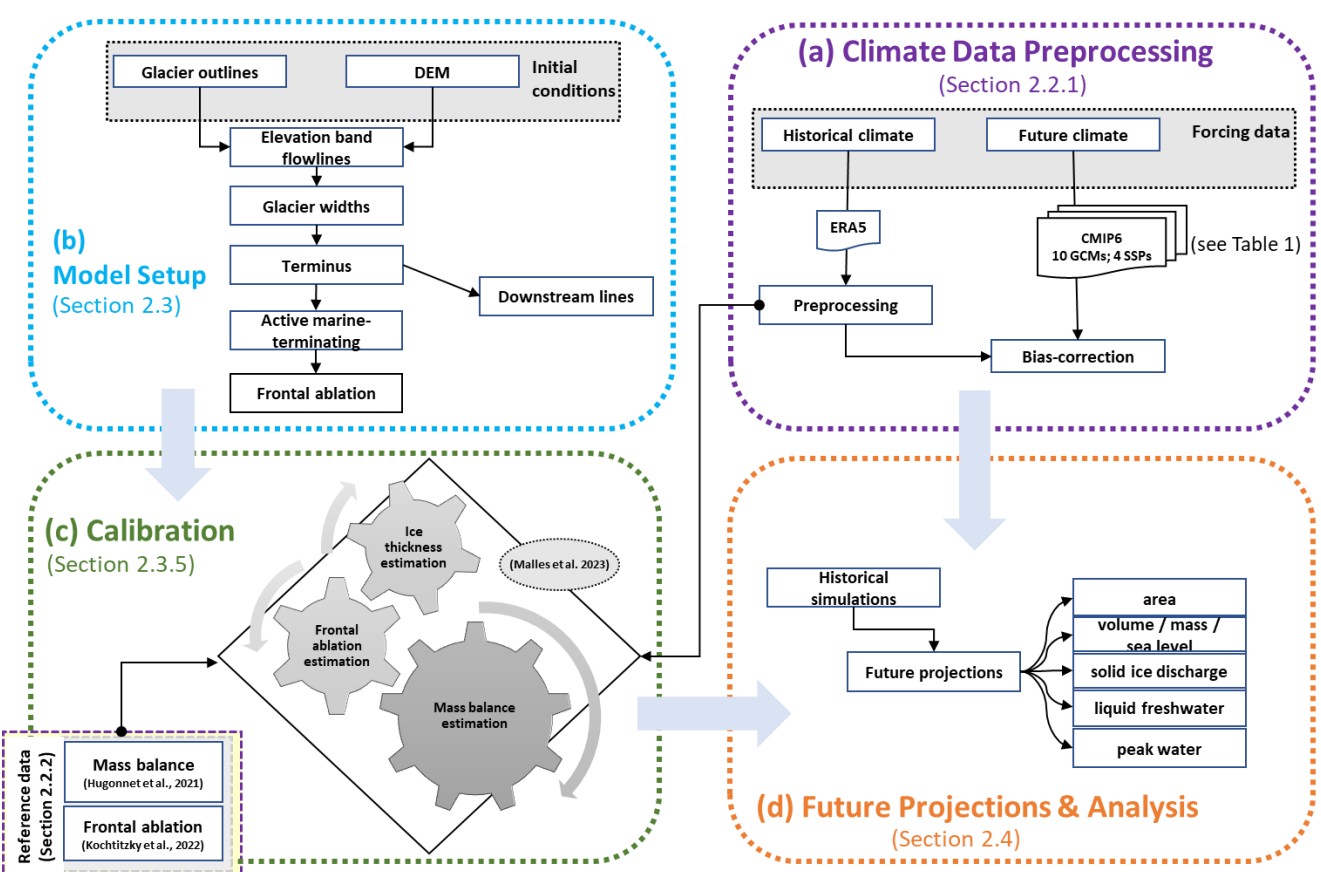

**Figure 2. Workflow of the Open Global Glacier Model (OGGM) applied in this study. The workflow comprises four main components: (a) Climate data preprocessing (Section 2.2.1), including the preparation of historical (ERA5) and future (CMIP6; 10 GCMs, 4 SSPs) climate forcing and subsequent bias correction; (b) Model setup (Section 2.3), which initializes glacier geometry using outlines, DEM, and initial conditions, and includes computation of frontal ablation; (c) Calibration (Section 2.3.5), which iteratively estimates ice thickness, frontal ablation, and mass balance following Malles et al. (2023), using reference datasets from Hugonnet et al. (2021) and Kochitzky et al. (2022); and (d) Future projections and analysis (Section 2.4), which simulate glacier evolution and freshwater contributions under bias-corrected CMIP6 scenarios.**

### 2.3.2 Mass Balance Model

The climate data is interpolated to the glacier location to compute the glacier's monthly surface mass balance. The air temperature data is corrected using the lapse rate described in Section 2.2.1. This calculation is performed at each grid point along the flowline of the glacier. The solid precipitation is calculated using a threshold air temperature. Specifically, all precipitation is considered solid when the air temperature is below 0°C. All precipitation is considered liquid when the air temperature is above 2°C. For temperatures between 0°C and 2°C, a linear interpolation between solid and liquid precipitation is applied. The monthly surface mass balance of a glacier, pertaining to the grid point $i$ located at elevation $z_i$, is computed for every grid point along the flowline.

$$m_i(z) = f_P P_i^{solid}(z) - \mu max(T_i^m(z), 0),$$ (4)

where $m_i(z)$ is monthly surface mass balance for grid point $i$ (in mm w.e.); $f_P$ is precipitation correction factor; $P_i^{solid}(z)$ is solid precipitation (in mm w.e.); $\mu$ is air temperature sensitivity (in mm w.e./K); $T_i^m(z)$ is the air temperature above the threshold for ice melt at the glacier surface (in K).

The precipitation factor ($f_P = 1.6$) was not calibrated in this study but adopted from the standard OGGM v1.4 framework, originally calibrated by Maussion et al. (2019) against the World Glacier Monitoring Service (WGMS) reference glaciers through extensive cross-validation. This global multiplicative correction accounts for orographic precipitation enhancement, snow redistribution through avalanches and wind-blown snow, and systematic underestimation in ERA5 reanalysis data. The precipitation factor represents a pre-determined global value from the OGGM framework that has been previously calibrated and validated for Arctic glacier applications (Maussion et al., 2019; OGGM Cross-Validation Dataset, 2024). To evaluate the performance of the global precipitation correction factor for regional Greenland peripheral glaciers, we compared scaled ERA5 precipitation against high-resolution Northeast Greenland Ice Stream Weather Research and Forecasting (NEGIS_WRF) model output (5 km resolution) covering the FIIC region for 2014-2018 (Turton et al., 2020). The comparison shows reasonable agreement (Supplementary Fig. S1 and Table S1), with ERA5 exhibiting a modest underestimation of 7 % (mean bias of 6.4 mm per month, median relative difference of 11.8 %) and moderate spatial-temporal correlation (r = 0.57). While the global parameter shows systematic underestimation, the magnitude is acceptable given that our glacier-specific calibration approach adjusts the temperature sensitivity parameter μ (Section 2.3.5) using observed geodetic mass balance, effectively compensating for any residual precipitation biases in total mass balance. This validation confirms that $f_P = 1.6$ provides adequate precipitation representation for our modeling framework, supporting its continued use for consistency with the broader OGGM community and enabling direct comparison with global glacier projections.

### 2.3.3 Enhanced Modeling of Marine-Terminating Glaciers

Accurately modeling marine-terminating glaciers is crucial for understanding their dynamics and predicting their response to climate change. In this study, we apply an enhanced approach by incorporating a module that accounts for hydrostatic pressure balance, enabling the SIA for marine-terminating glaciers with terminal cliffs (Malles et al., 2023). In the enhanced parametrization, the sliding velocity calculation was also updated to take the water depth of the glacier's bed into account. The sliding velocity calculation considers the height above buoyancy, calculated as the difference between ice thickness and the ratio of ice and ocean water densities multiplied by water depth. OGGM was updated for consistency in the dynamical model core and ice thickness inversion, incorporating height above buoyancy and frontal ablation parameterization. The same frontal ablation parameterization is applied in the dynamical model, ensuring a consistent ice thickness inversion solution for all glaciers. The parameterized frontal ablation flux is subtracted from the flux through the grounding line in every time step. When the accumulated difference is sufficiently positive/negative, the glacier can advance/retreat into the next grid cell. If the thickness of one or more grid cells falls below flotation in a specific time step, the part of this volume that is contained in grid cells beyond the one adjacent to the last grid cell above flotation is removed and added to the frontal ablation output variable (i.e., the formation of ice shelves is suppressed).

Frontal ablation ($Q_f$) in marine-terminating glaciers is determined by employing the calculation method proposed by Oerlemans and Nick (2005):

$$Q_f = kdhw, \tag{5}$$

Where $k$, $d$, $h$, and $w$ are water-depth sensitivity parameter (in $\text{yr}^{-1}$), water depth (in m), ice thickness (in m), and width at the glacier front (in m), respectively. An iterative procedure is employed to find a value for the water-depth sensitivity parameter that produces a frontal ablation estimate within the uncertainty bounds of the data used. This value is used in ice thickness inversion and a subsequent historical dynamical run. The mass loss through frontal ablation is considered as solid ice discharge. For a more detailed description of this process, including its implementation in OGGM, readers are referred to Malles et al. (2023).

### 2.3.4 Freshwater Runoff and Peak Water Calculations

All the runoff generated through surface melt processes and direct rain is considered as liquid freshwater runoff. The total annual freshwater runoff from the glacier, following the fixed-gauge approach, where runoff includes all sources within the original glacier boundaries, was calculated by summing the components of off-glacier snowmelt, on-glacier melt, on-glacier liquid precipitation, and off-glacier liquid precipitation.

$$TR = \sum GR_{i,s,r} + SR + RR, \tag{6}$$

Where $TR$ is total liquid freshwater runoff, $GR_{i,s,r}$ denotes the sum of runoff from glacier ice ($GR_i$), snow ($GR_s$), and rain ($GR_r$), $SR$ is snowmelt off-glacier, and $RR$ is rain runoff off-glacier. $SR$ and $RR$ are the freshwater runoff components from the deglaciated areas within the RGI boundaries, where the glacier has retreated or disappeared over time. Although the glaciers have retreated from these areas, they still contribute to the total freshwater runoff due to initial boundary constraints and are therefore included in the calculation.

This study employs the glacier-centric "fixed-gauge" approach standard in glacier hydrology studies (Bliss et al., 2014; Huss and Hock, 2018; Jansson et al., 2003; Wimberly et al., 2025; Rounce et al., 2023; Zekollari et al., 2025). This methodology tracks runoff from areas defined by initial glacier boundaries as glaciers retreat, enabling isolation of glacier-specific hydrological changes and direct comparison with established global glacier literature. The fixed-gauge approach is scientifically meaningful because it captures the complete hydrological contribution from areas that were initially glaciated, allowing assessment of how water yield from these specific areas changes as glaciers retreat while maintaining consistency with established glacier mass balance and runoff studies that form the basis for water resource planning in glacier-fed basins. While a catchment-based approach would provide complementary insights into total watershed hydrology, it addresses a fundamentally different research question and is technically incompatible with OGGM's design. OGGM is explicitly designed as a glacier-centric model that operates on individual glaciers as the smallest dynamically independent entity (Maussion et al., 2019). The model's ice dynamics module computes ice flux along individual flowlines and cannot handle the complex multi-

glacier boundaries that catchments would introduce. In Greenland, catchments frequently contain multiple peripheral glaciers plus portions of the main ice sheet, a configuration that would violate OGGM's fundamental assumption of single-glacier ice divides. Since OGGM is designed specifically for peripheral glaciers and does not include ice sheet dynamics, implementing a catchment-based approach would require coupling with an ice sheet model, which is beyond the scope of this study and our modeling framework's capabilities. Our glacier-focused approach specifically addresses glacier response to climate change rather than general catchment hydrology, which aligns with our research objectives and enables direct comparison with the established literature.

Technically, OGGM is not capable of calculating catchment runoff outside glacier boundaries, limiting our analysis to glacier-defined areas. However, our approach ensures that coupling to hydrological models is possible (Hanus et al., 2024), which represents the ideal solution for comprehensive catchment analysis. The alternative - having the glacier model cover only the current glacier extent - would require hydrological models to operate over time-dependent domains as glaciers retreat, which presents significant technical challenges.

"Peak water" is defined as the moment in time when the amount of annual freshwater released from a glacier reaches its highest level and begins to decrease. As a glacier shrinks, more annual meltwater is released until a maximum is reached. This represents "glacier peak water", a well-established glaciological concept defined as maximum annual runoff from initially glaciated areas (Bliss et al., 2014; Huss and Hock, 2018), which is distinct from "catchment peak water" and directly relevant for understanding glacier response to climate change. Peak water is determined after applying an 11-year rolling mean to the total liquid freshwater runoff time series to reduce short-term variability and highlight long-term trends.

### 2.3.5 Model Calibration

In previous versions of OGGM, spatial interpolation was used in the calibration process of the surface mass balance model due to the lack of observational data. However, we are now able to calibrate on a glacier-by-glacier basis using geodetic mass balance (Hugonnet et al., 2021) and frontal ablation data, including volume changes below sea level (Kochtitzky et al., 2022). We use the following equation after Malles et al. (2023) for calibration of the air temperature sensitivity μ:

$$\mu = \left( f_p P_{solid} - \frac{\Delta M_{awl} + C + f_{bwl}\Delta M_f}{A_{rgi}} \right) \frac{1}{T_m}, \tag{7}$$

Where $\Delta M_{awl}$ is observed annual volume change above sea level of a glacier (Gt/yr) as given by Hugonnet et al. (2021), $C$ is observed annual frontal ablation rate of a glacier as given by Kochtitzky et al. (2022) (Gt/yr), $\Delta M_f$ is observed annual volume retreat due to area changes in the terminus region of a glacier (Gt/yr), as given by Kochtitzky et al. (2022), $f_{bwl}$ is an assumed fraction of $\Delta M_f$ occurring below the waterline, $A_{RGI}$ is glacier surface area of a glacier as given by the RGI 6.0 (km$^2$), $T_m$ is annually accumulated air temperature (K) above the threshold for ice melt (-1 °C) at the glacier surface. For a comprehensive description of the calibration process, readers are referred to Malles et al. (2023). A comprehensive summary of all model parameters, their values, and calibration methods is provided in Supplementary Table S2.

**2.4 Statistical Analysis**

Finally, future glacier area, volume, mass loss, sea level rise, solid ice, freshwater runoff contributions, and peak water were
300 projected from 2020 to 2100 for all peripheral glaciers in Greenland. We employed several tests to analyze the data and assess
the statistical significance of our findings. Statistical significance is defined as $p < 0.05$ throughout this study. One-way
Analysis of Variance (ANOVA) was used to compare means across multiple groups (e.g., emission scenarios) for normally
distributed data (Fisher, 1992). Two-way ANOVA examined the effects of two independent variables (e.g., region and
emission scenario) on a dependent variable, as well as their potential interaction. The F-statistic in ANOVA, representing the
305 ratio of between-group variability (variation between sample means) to within-group variability (variation between sample
means), was used to quantify the significance of differences. For non-normally distributed data, we used the Kruskal-Wallis
test, a non-parametric alternative to one-way ANOVA (Kruskal and Wallis, 1952). Following significant results, Tukey's
Honestly Significant Difference (HSD) test was applied for post-hoc pairwise comparisons (Tukey, 1949). These methods
assessed differences in glacier area retreat, volume loss, sea level rise contributions, freshwater runoff, and peak water timing
across emission scenarios and regions. The choice of test depended on data characteristics and comparison specifics especially
the data distribution.

**3 Results**

**3.1 Projected Glacier Area Retreat, Volume Loss, and Sea Level Rise Contributions**

Our projections suggest notable declines in area and volume of glaciers along the periphery of Greenland by the year 2100
across all evaluated emission scenarios (see Figs. 3 and 4). A one-way ANOVA test revealed significant differences in area
retreat among SSP scenarios, indicating the varied impacts of emission levels on the spatial changes of Greenland's peripheral
glaciers.

Under the low-emission scenario (SSP126), glacier area shows a relatively steady annual decrease of $0.18 \pm 0.03$ %/yr (mean
$\pm 1$ SD), in contrast to the high-emission scenario (SSP585), which exhibits a more pronounced annual decline of $0.43 \pm 0.08$
320 %/yr (Fig. 3b). Additionally, a trend towards increasing standard deviation over time across all scenarios indicates growing
variability in the projections of the remaining glacier area, reflecting increased uncertainty as the century progresses.
Projections suggest a decrease in total glacier area of $19 \pm 6$ % under SSP126 and $44 \pm 15$ % under SSP585 by 2100. Regional
patterns show pronounced differences: North retains $63 \pm 10$ % (SSP126) to $42 \pm 15$ % (SSP585) of area, North-East $74 \pm 7$
% to $54 \pm 12$ %, Central-East $58 \pm 11$ % to $28 \pm 13$ %, Central-West $44 \pm 9$ % to $21 \pm 10$ %, South-East $73 \pm 8$ % to $49 \pm 14$
325 %, and South-West $70 \pm 9$ % to $48 \pm 13$ % (Fig. 3b). FIIC demonstrates exceptional stability, maintaining > 95 % of its area
across all scenarios through 2100 (Fig. 3a)

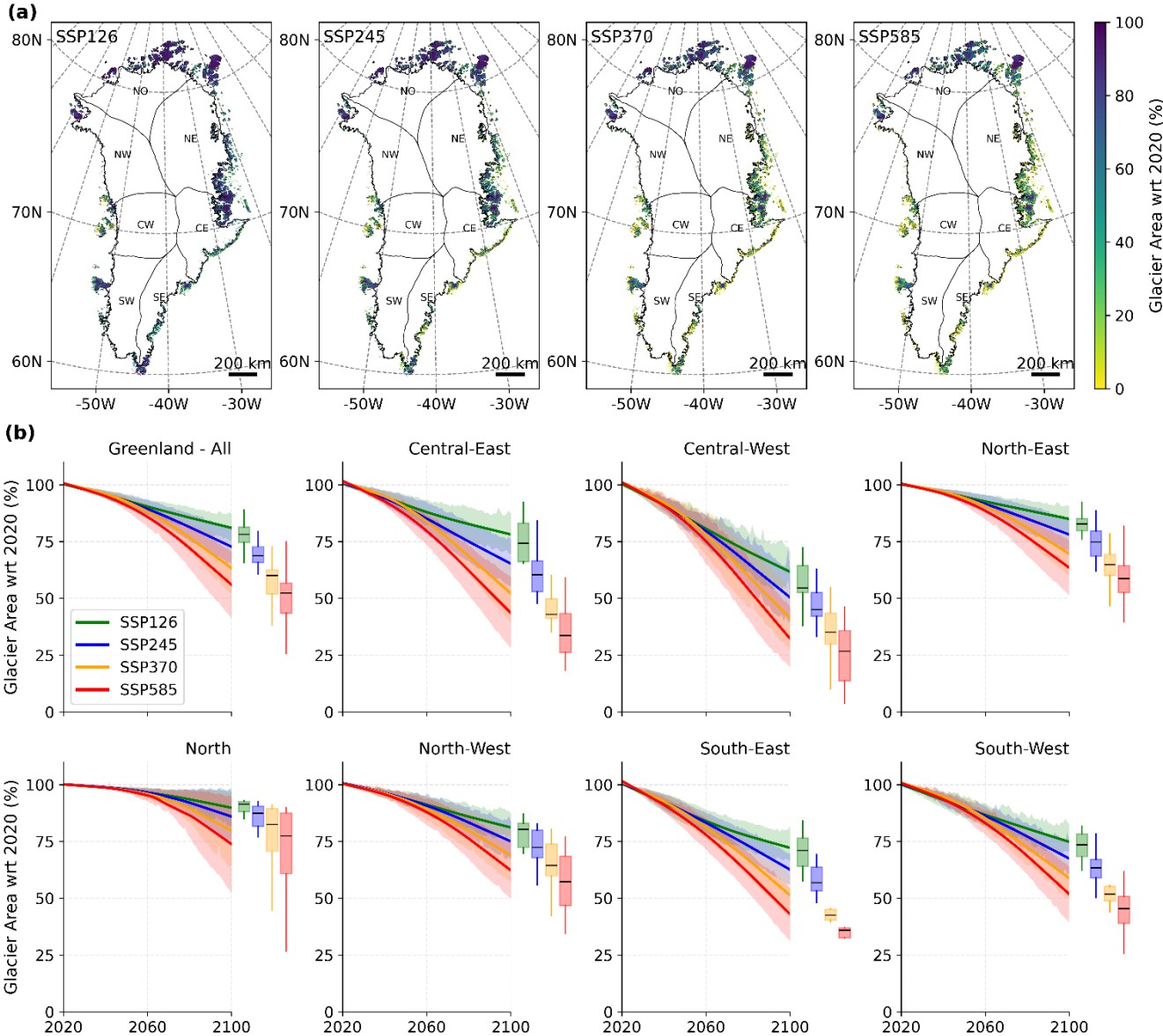

**Figure 3. Projected evolution of Greenland's peripheral glacier area under different emission scenarios. (a)** Spatial distribution of the projected remaining glacier area in 2100 relative to 2020, averaged across 10 CMIP6 GCMs for each SSP. The glaciers are represented as polygons from RGI 6.0 outlines. **(b)** Temporal evolution of the glacier area from 2020 to 2100 (mean ± 1 SD). Solid lines represent the ensemble mean smoothed using the locally estimated scatterplot smoothing (LOESS) method, and shaded regions denote the inter-model spread (± 1 SD). Box plots show the distribution of projected glacier area in 2100 relative to 2020 across the four SSP scenarios (10 GCMs).

Similarly, glacier volume is expected to decrease by 29 ± 6 % under SSP126 and 52 ± 14 % under SSP585, with a significant regional variability (Fig. 4). For instance, the Central-West subregion is projected to experience the most severe volume loss of 56 ± 9 % under SSP126 and 79 ± 10 % under SSP585, which is statistically higher than other regions. Conversely, the North-East region shows the lowest projected loss of 22 ± 4 % under SSP126 and 39 ± 9 % under SSP585 (Fig. 4b). A two-

way ANOVA confirms that both the subregion and SSP scenario have a significant impact on the projected glacier volume loss, independent of each other. However, no interaction effect was observed between region and SSP, indicating that the impact of SSP on projected total volume loss does not significantly differ across regions and vice versa.

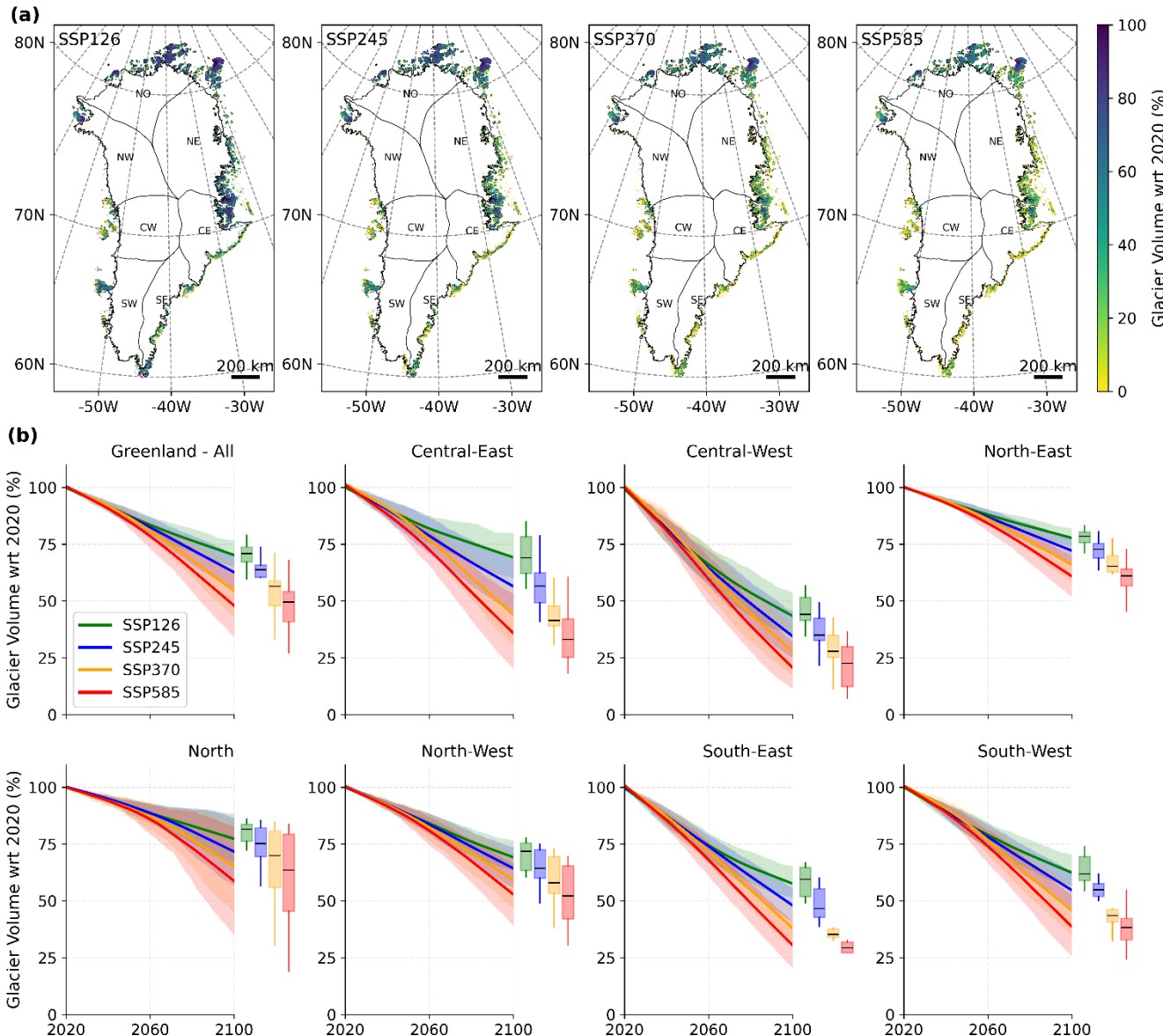

**Figure 4. Projected evolution of Greenland's peripheral glacier volume under different emission scenarios. (a)** Spatial distribution of the projected remaining glacier volume in 2100 relative to 2020, averaged across 10 CMIP6 GCMs for each SSP. The glaciers are represented as polygons from RGI 6.0 outlines. **(b)** Temporal evolution of glacier volume from 2020 to 2100 (mean ± 1 SD). Solid lines represent the ensemble mean smoothed using the locally estimated scatterplot smoothing (LOESS) method, and shaded regions indicate the inter-model spread (± 1 SD). Box plots show the distribution of projected glacier volume in 2100 relative to 2020 for the four SSP scenarios (10 GCMs).

Regional variability in glacier response is strongly influenced by glacier size and elevation distributions (Supplementary Figs. S2-S18). Under SSP585, small glaciers (< 1 km²) at low elevations (< 500 m) lose 85-95 % of their area by 2100 across most subregions, while larger glaciers (> 10 km²) at higher elevations (> 1000 m) retain 40-60 % of their area in North-East but only 10-30 % in Central-West. The North-East region benefits from its favorable elevation distribution, with 45 % of glacier area concentrated above 1000 m elevation (Supplementary Fig. S2), providing substantial accumulation areas that buffer against warming. In contrast, Central-West has 70 % of its glacier area below 800 m elevation, making these glaciers highly vulnerable to atmospheric warming. Central-East shows the strongest emission-dependent divergence: glaciers at 600-800 m elevation retain 45-55 % of volume under SSP126 but only 15-25 % under SSP585 by 2100 (Supplementary Figs. S7, S15), reflecting this region's proximity to critical thermal thresholds where modest warming differences trigger disproportionate responses.

The losses in glacier volume translate to a contribution to sea level rise of $10 \pm 2$ mm under SSP126 and $19 \pm 5$ mm under SSP585, with substantial regional variability (Fig. 5a). For all SSPs, sea level rise (SLR) shows significant positive trends over 2021 to 2100: SSP126 ($+0.10 \pm 0.01$ mm/yr), SSP245 ($+0.13 \pm 0.02$ mm/yr), SSP370 ($+0.16 \pm 0.03$ mm/yr), and SSP585 ($+0.19 \pm 0.04$ mm/yr) (Fig. 5a). The North-East subregion exhibits the strongest acceleration in SLR contribution ($+0.092 \pm 0.027$ mm/yr², representing the rate of change of the annual SLR trend) and the highest mean SLR contribution by 2100 across all SSPs. Under SSP585, it is projected to contribute 37 % of the total SLR (Fig. 5b). In contrast, the Central-West subregion is suggested to have the weakest acceleration ($+0.0082 \pm 0.0015$ mm/yr²) and the lowest projected SLR contribution (3 %) under SSP585.

 Size-dependent responses contribute to regional SLR patterns. In North-East, large glaciers (> 10 km²) above 1000 m elevation contribute 65 % of the region's SLR despite comprising only 30 % of glacier count (Supplementary Figs. S5, S13), reflecting sustained mass loss from substantial ice volumes at high elevations. Conversely, Central-West's SLR contribution is dominated by rapid depletion of numerous small glaciers (< 1 km²) at low elevations (< 600 m), which lose 85-90 % of volume by 2100 but contribute proportionally less to SLR due to limited initial ice mass.

A one-way ANOVA highlighted significant differences in mean SLR contributions between subregions for each SSP. Additionally, two-way ANOVA analysis underscored the significant interaction between subregions and emission scenarios on end-of-century area, volume losses, and SLR contributions, demonstrating the compound influence of local environmental factors and global emission trajectories on the dynamics of glacier evolution. These findings indicate that SLR from Greenland's peripheral glaciers will substantially increase through the 21st century under all SSPs.

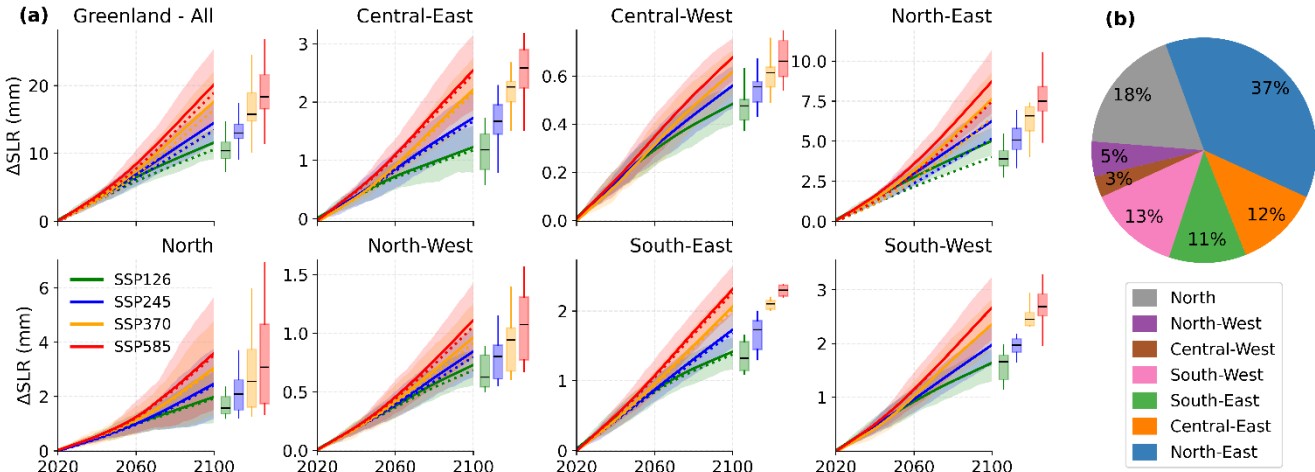

**Figure 5. Projected cumulative mass loss of Greenland's peripheral glaciers and corresponding sea-level rise contributions. (a) Solid lines show the projected cumulative mass change (mean ± 1 SD) in sea-level equivalent (SLE, mm) from 2020 to 2100 across different subregions and emission scenarios (10 GCMs). Dotted lines indicate the sea-level rise component considering only the mass loss below sea level. The ensemble means are smoothed using the locally estimated scatterplot smoothing (LOESS) method, with shaded areas representing the inter-model spread (± 1 SD). Box plots show the interquartile range of cumulative SLE contributions by 2100 across subregions. (b) Relative contributions (%) of each subregion to total sea-level rise by 2100 under SSP5-8.5 (mean of 10 GCMs). The order of subregions in the pie chart follows their approximate geographic position around Greenland periphery.**

### 3.2 Freshwater Contributions: Solid Ice Discharge vs Liquid Freshwater Runoff

Our projections reveal significant but contrasting trends in both solid ice discharge and liquid freshwater runoff from Greenland's peripheral glaciers over the 21st century, influenced by climate change and emission scenarios. In general, the freshwater runoff clearly is the dominant term of mass loss compared to solid ice discharge.

Solid ice discharge shows an average of $3.0 \pm 0.7$ Gt/yr from 2020 to 2100 under the high-emission SSP585 scenario, with a notable decrease post-2050 attributed to the diminishing extent of marine-terminating glaciers (Fig. 6). Accordingly, the solid ice discharge exhibits a declining trend under all scenarios, with substantial interannual variability. For example, under SSP126, the solid ice discharge shows accelerating decline at -0.011 Gt/yr² (acceleration in the rate of decrease), significant accelerating decline mirrored across other scenarios: SSP245 (-0.014 Gt/yr²), SSP370 (-0.017 Gt/yr²), and SSP585 (-0.018 Gt/yr²).

In terms of regional ice discharge, most areas exhibit declining trends, except for the North-East, which shows a marginal increase from 1.05-1.06 Gt/yr in 2021-2030 to 1.15-1.23 Gt/yr by 2091-2100 under low and high emission scenarios. Two-way ANOVA tests confirm significant differences in ice discharge between the period I (2021-2030) and period II (2091-2100) of projections. However, no significant differences are found among emission scenarios or in the interaction between scenarios and selected decades. It is important to note that our model does not account for ocean temperature changes, which may affect solid ice discharge projections.

The regional differences in solid ice discharge are strongly influenced by the distribution and evolution of marine-terminating glaciers across Greenland's periphery (Fig. 6b). In 2020, Greenland's peripheral glaciers included 405 marine-terminating

glaciers covering 20,248 km². The distribution is highly heterogeneous: South-East has the highest count (139 glaciers, 34.3 % of total) covering 5,172 km² (25.5 %), followed by Central-East (126 glaciers, 31.1 %; 2,999 km², 14.8 %) and North (66 glaciers, 16.3 %; 6,112 km², 30.2 %). The North-East contains 47 marine-terminating glaciers (11.6 %) covering 2,792 km²

(13.8 %), while Central-West and South-West have minimal marine-terminating coverage (4 and 5 glaciers respectively, together 2.2 % of total). Our projections reveal substantial reductions in marine-terminating glacier extent by 2100, with considerable regional variability (Fig. 6b). Under SSP585, Greenland-wide marine-terminating glacier count declines to 18 ± 6 % of 2020 levels by 2100, with area declining to 20 ± 8 %. Regional patterns vary markedly: Central-West's marine-terminating glaciers essentially disappear by mid-century across all scenarios, while North-East demonstrates greater

resilience, retaining 65 ± 12 % (SSP126) to 35 ± 10 % (SSP585) of marine-terminating area by century's end. The rate of transition varies significantly between scenarios, with Central-East showing particularly strong divergence, retaining 25 ± 9 % (SSP126) versus only 10 ± 5 % (SSP585) of marine-terminating area by 2100.

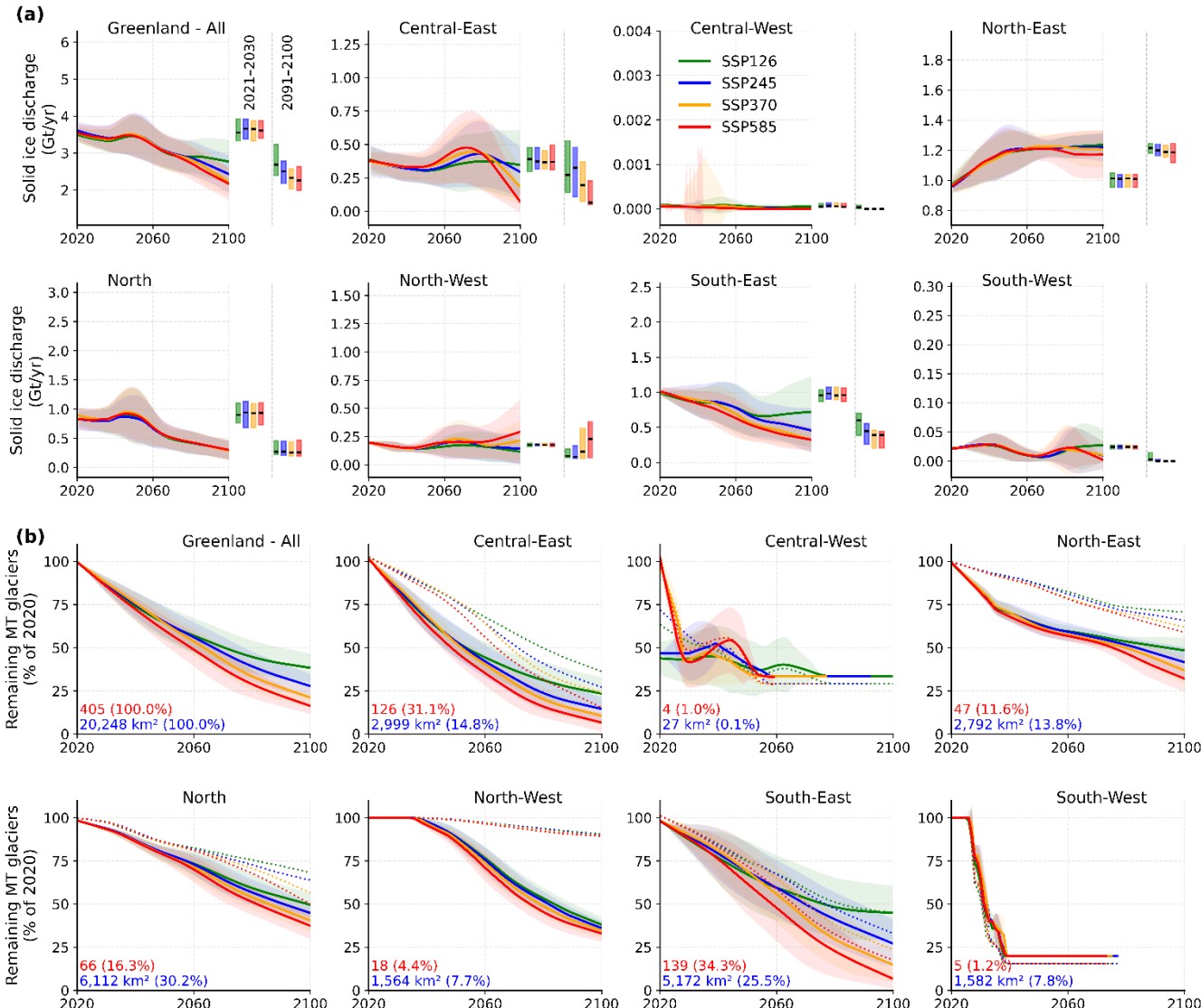

**Figure 6. Projected solid ice discharge and evolution of marine-terminating glaciers under different emission scenarios. (a) Solid ice discharge (Gt/yr) from Greenland's peripheral glaciers and individual subregions from 2020 to 2100 (mean ± 1 SD; 10 GCMs). The solid lines represent ensemble means smoothed using the locally estimated scatterplot smoothing (LOESS) method, and the shaded areas denote the inter-model spread (± 1 SD). The first set of box plots represents the average solid ice discharge over the first decade of projections (2021–2030), and the second set represents the average over the last decade of projections (2091–2100) under different SSPs. (b) Projected percentage of remaining marine-terminating (MT) glaciers (count: solid lines and area: dotted lines) relative to 2020. The numbers in red denote the count of MT glaciers (and % of total MT glaciers) with volume below sea level in 2020. The numbers in blue denote the area of MT glaciers (and % of total MT glaciers area) with volume below sea level in 2020. The solid lines and shaded bands indicate the ensemble mean (± 1 SD) across the four SSP scenarios (10 GCMs).**

Conversely, projections for liquid freshwater runoff indicate a significant increase over the century, with annual averages ranging from 138 ± 12 Gt/yr under SSP126 to 184 ± 27 Gt/yr under SSP585 (Fig. 7a). Freshwater runoff increases under SSP585 from 145 ± 27 Gt/yr to 216 ± 46 Gt/yr, whereas it decreases under SSP126 from 145 ± 25 Gt/yr to 120 Gt/yr by 2100

compared to 2020. The North-East subregion emerges as the dominant contributor, accounting for 35 % of the total runoff over 2020-2100 under SSP585. This contribution is contrasted sharply by the Central-West region, which contributes only 3 % of the total annual freshwater runoff (Fig. 7b). These regional differences in runoff contributions are influenced by variations in glacier number, area, and ice volume among subregions (Fig. 1c-d). The spatial heterogeneity in runoff contributions reflects

the combined effects of initial glacier distribution and differential response to warming. North-East's 35 % contribution stems from its large initial ice volume (37 % of total) and sustained high-elevation accumulation areas that continue producing substantial meltwater throughout the century. Central-West's minimal 3 % contribution reflects both its small initial glacier coverage (5 % of total area) and severe volume depletion, with 79 ± 10 % volume loss by 2100 under SSP585 leaving limited ice mass to generate runoff.

The liquid freshwater runoff is the dominant mass loss component throughout the century, with annual averages approximately 45 to 60 times larger than solid ice discharge across all scenarios. This dominance of liquid freshwater runoff over solid ice discharge has important implications for understanding glacier mass loss pathways. While solid ice discharge shows consistent declining trends (-0.011 to -0.018 Gt/yr²), freshwater runoff increases substantially until peak water is reached, after which it begins to decline under lower emission scenarios but continues to increase under higher emission scenarios through 2100. This

divergence means that liquid freshwater becomes increasingly dominant for understanding glacier impacts on fjord systems and sea level rise.

The temporal evolution of these two freshwater components reflects the transition of Greenland's peripheral glaciers from systems with significant marine-terminating components to predominantly land-terminating, surface-melt-dominated systems. In 2021-2030, solid ice discharge accounts for 2.1 ± 0.4 % of total freshwater contributions under SSP585, declining to 1.3 ±

0.3 % by 2091-2100 as marine-terminating glaciers retreat inland (Fig. 6b). This temporal shift is particularly pronounced in regions with extensive marine-terminating coverage: North-East maintains the highest absolute solid ice discharge throughout the century (1.05-1.23 Gt/yr), though even here it represents only 1-2 % of total freshwater output. Central-East, despite having the second-highest marine-terminating glacier count initially, experiences one of the steepest declines in both absolute and relative solid ice discharge contribution, with values decreasing from 0.40 Gt/yr (2021-2030) to 0.22 Gt/yr (2091-2100) under

SSP585 as most marine-terminating glaciers transition to land-terminating positions by mid-century.

The composition of freshwater runoff is also expected to shift markedly over the century. Under SSP585, the proportion of glacier meltwater in total runoff is projected to decrease from 92 % in 2021-2030 to 72 % by 2091-2100. Meanwhile, contributions from off-glacier rainfall and snowmelt are expected to increase from less than 1 % to 8 % (approximately 8-fold) and from 1 % to 15 % (approximately 15-fold), respectively (Fig. 7c-d). The seasonal distribution of freshwater runoff

components is also projected to change significantly (Fig. 7c-d). In 2021-2030, glacier melt dominates runoff from May to September, peaking in July. By 2091-2100, while glacier melt still peaks in July, its contribution is notably reduced. Snowmelt shows a marked increase, especially in May-June, while rainfall contributions increase throughout the year especially during summer months.

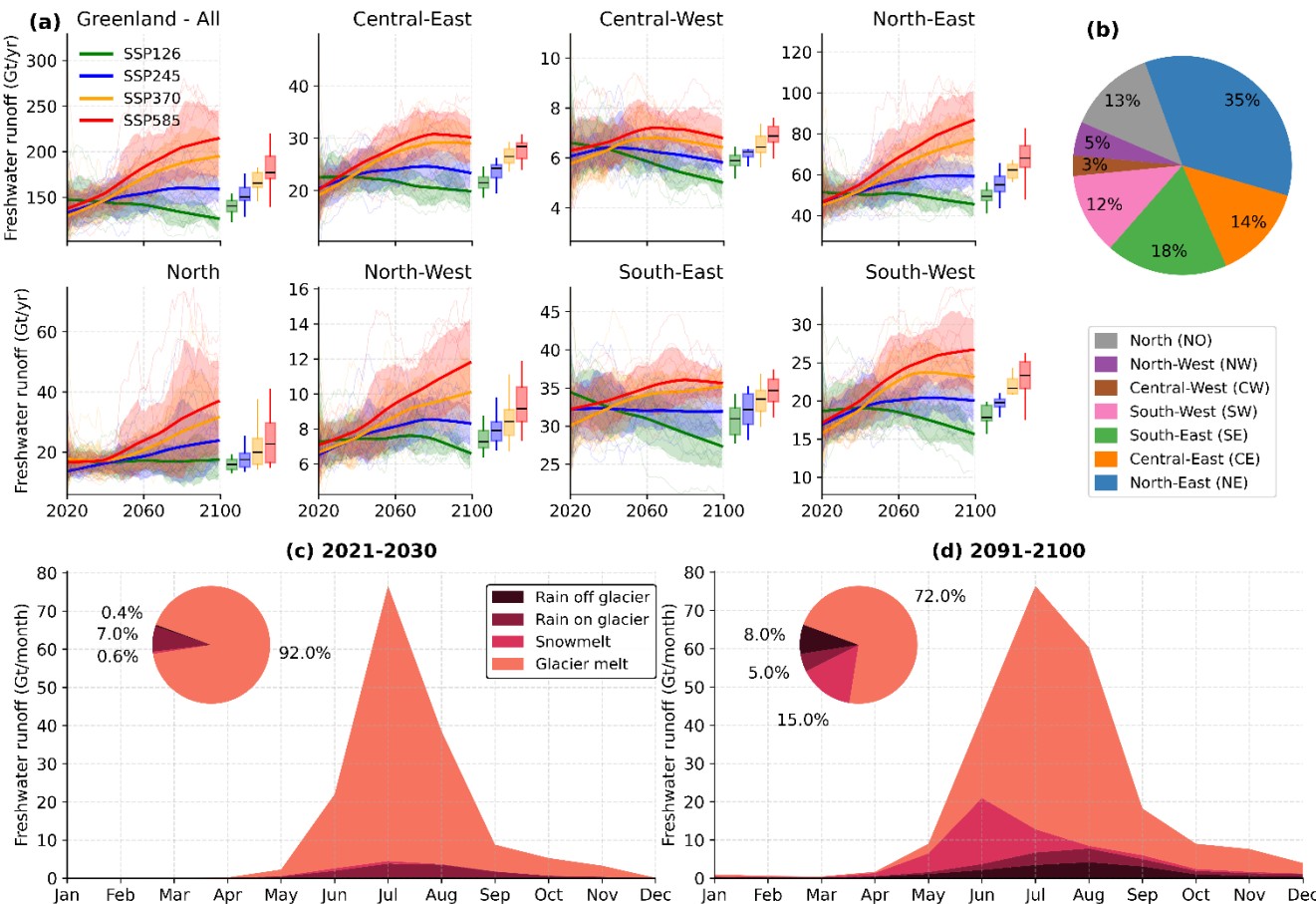

Figure 7: Figure 7. Projected freshwater runoff from Greenland's peripheral glaciers and its components under different emission scenarios. (a) Freshwater runoff (Gt/yr) from individual subregions during 2020–2099 under four SSP scenarios (10 GCMs). Solid lines represent ensemble means smoothed using the locally estimated scatterplot smoothing (LOESS) method, with shaded areas denoting the inter-model spread (± 1 SD). Box plots show the statistics of average freshwater runoff over the projection period. (b) Percent contributions of freshwater runoff from different subregions under SSP585. The order of subregions in the pie chart follow their approximate geographic position around Greenland periphery. (c-d) Seasonal distribution (Gt/month) and average percent contributions of individual runoff components (glacier melt, snowmelt, rain on glacier, and rain off glacier) to total freshwater runoff during the first decade of projections (2021-2030) and the last decade of projections (2091-2100) under SSP585.

### 3.3 Peak Water Timing and Magnitude

The timing and magnitude of peak water runoff from Greenland's peripheral glaciers are significantly influenced by varying emission scenarios, demonstrating notable spatial and temporal variability (Fig. 8).

For Greenland peripheral glaciers, peak water runoff is projected to occur around the year 2050 ± 21 under the low-emission SSP126 and around 2082 ± 9 under the high-emission SSP585 scenario (Fig. 8a). The shift of nearly 30 years is statistically significant (Kruskal-Wallis $p < 0.05$), indicating a strong influence of emission scenarios on the hydrological responses of the glaciers. The maximum runoff at these peak times is expected to be 214 ± 21 Gt/yr under SSP126 and 293 ± 61 Gt/yr under SSP585 (Fig. 8b), underscoring the increased runoff associated with higher emissions.

Subregional analysis reveals that southern regions such as South-East and South-West are expected to experience earlier peak waters, with median timings around 2038 (± 17 years) and 2035 (± 10 years) under SSP126, respectively. Conversely, northern and central subregions show a delayed response; for instance, the North-East and North regions are projected to reach their peak around 2053 (± 22 years) and 2055 (± 25 years) under SSP126, shifting to 2080 (± 19 years) and 2086 (± 13 years) under

SSP585 (Fig. 8a). The non-monotonic pattern in the South-East region (SSP126: 2038, SSP245: 2050, SSP370: 2042, SSP585: 2055) highlights the complex, non-linear relationship between warming scenarios and peak water timing in glacier systems. Under moderate warming (SSP126), glaciers experience enhanced melt that quickly peaks as they approach a new, smaller equilibrium state relatively early. SSP245's intermediate warming prolongs the melt enhancement phase, delaying peak water as glaciers take longer to stabilize. SSP370's more aggressive warming accelerates glacier response, causing earlier exhaustion

of melt potential compared to SSP245, while SSP585's extreme warming sustains high melt rates for an extended period by continuously accessing deeper ice reserves until substantial glacier depletion occurs. Despite these apparent differences in timing across subregions, statistical analysis using the Kruskal-Wallis test indicates that these variations are not statistically significant, suggesting that while regional differences exist, they do not diverge significantly under different scenarios.

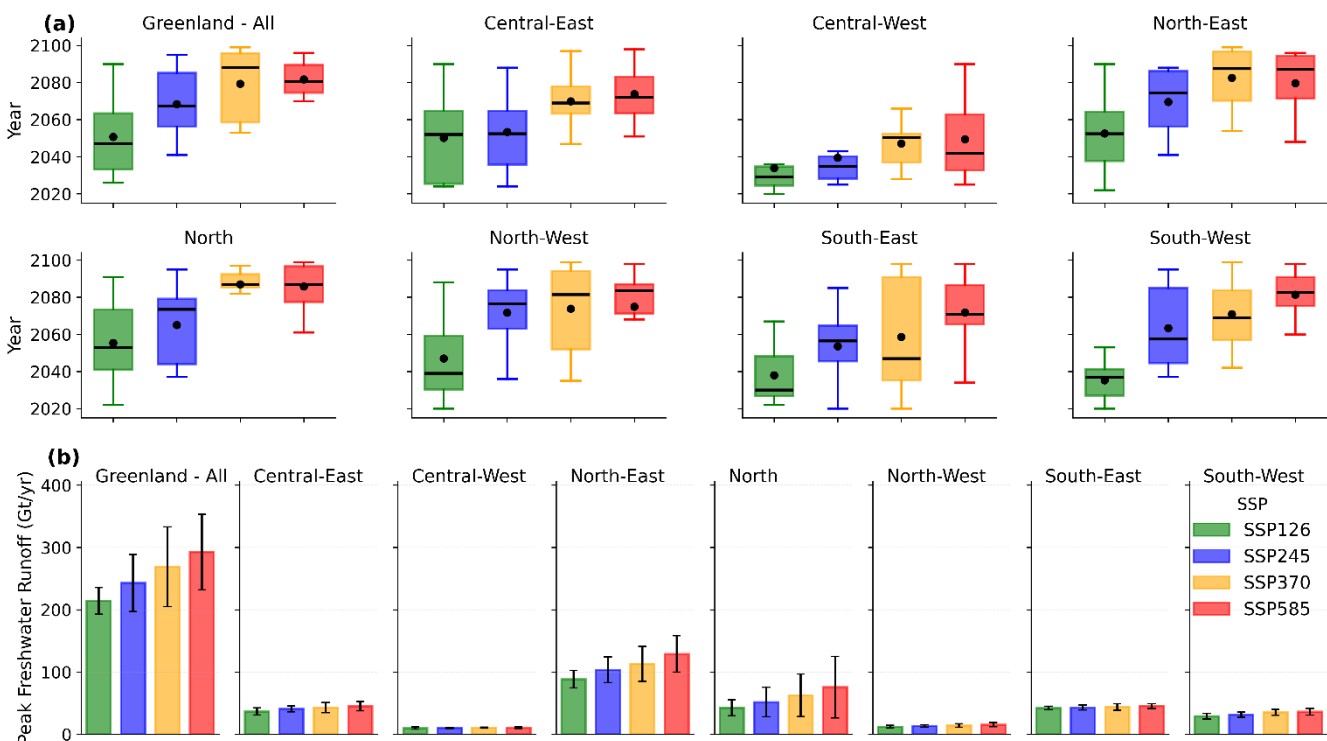

**Figure 8. Peak water timing and magnitude for Greenland's peripheral glaciers under different emission scenarios. (a) Box plots showing the distribution of peak water years across subregions under four SSP scenarios (10 GCMs). (b) Mean maximum freshwater runoff (Gt/yr) at the peak water year (mean ± 1 SD; 10 GCMs) for each subregion.**

## 4 Discussion

### 4.1 Increasing Glacier Mass Losses and Contribution to Sea Level Rise

Our projections indicate substantial losses in both area and volume of Greenland's peripheral glaciers by 2100, highlighting their high sensitivity to climatic changes. Under the high-emission scenario (SSP585), glacier area and volume are expected to decline by up to 44 % and 52 %, respectively, by 2100 (Fig. 3 & 4). For Greenland peripheral glaciers, these losses align with prior model based studies predicting accelerated glacier retreat and mass loss (up to 50 % by 2100) in response to warming air temperatures (Hock et al., 2019; Marzeion et al., 2020; Rounce et al., 2023).

To contextualize our findings within the broader landscape of glacier modeling studies, we compare our projections with recent global and regional glacier model outputs (Table 2). Our results show good agreement with other OGGM-based studies and fall within the range of multi-model ensemble projections, providing confidence in our modeling approach.

**Table 2.** Comparison of projected changes for Greenland's peripheral glaciers by 2100 across different modeling studies.

| Study | Model | Scenario | Volume/Mass loss (%) | Area loss (%) | SLR (mm) | Peak Runoff (Gt/yr) | Peakwater (year) |
|---|---|---|---|---|---|---|---|
| This study | OGGM v1.5.3 | SSP126 | 44 ± 15 | 19 ± 6 | 10 ± 2 | 214 ± 21 | 2050 ± 21 |
| | | SSP585 | 52 ± 14 | 44 ± 15 | 19 ± 5 | 293 ± 61 | 2082 ± 9 |
| Kang et al. (2024) | OGGM 1.6 | SSP126 | 48 ± 17 | 39 ± 18 | 19.5 ± 7 | 249 ± 52 | 2037 |
| | | SSP585 | 67 ± 18 | 61 ± 20 | 27.5 ± 7 | 298 ± 31 | 2083 |
| Zekollari et al. (2024) | OGGM 1.6 | SSP126 | 34 ± 21 | - | - | - | - |
| | | SSP585 | 55 ± 33 | - | - | - | - |
| | GloGEM | SSP126 | 26 ± 19 | - | - | - | - |
| | | SSP585 | 47 ± 31 | - | - | - | - |
| Rounce et al. (2023) | pyGEM | SSP126 | 33 ± 14 | - | 12 ± 6 | - | - |
| | | SSP585 | 50 ± 27 | - | 19 ± 11 | - | - |
| Marzeion et al. (2020) | Multi Models | SSP126 | 22 ± 24 | 21 ± 8 | 12 ± 5 | - | - |
| | | SSP585 | 42 ± 29 | 42 ± 18 | 22 ± 6 | - | - |

Our projected volume loss of 52 ± 14 % under SSP585 falls within the range of recent studies (42-67 %), with particularly
close agreement with Zekollari et al. (2024) using OGGM v1.6 (55 ± 33 %) and Rounce et al. (2023) using PyGEM (50 ± 27%). The slightly higher estimates from Kang et al. (2024) (67 ± 18 %) may reflect differences in frontal ablation parameterization or climate forcing details. Our area loss projections (44 ± 15 % under SSP585) similarly align well with multi-model estimates from Marzeion et al. (2020) (42 ± 29 %) and Kang et al. (2024) (61 ± 20 %).

For sea level rise contributions, our projection of 19 ± 5 mm under SSP585 is consistent with, though slightly lower than,
estimates from Kang et al. (2024) (27.5 ± 7 mm), Marzeion et al. (2020) (22 ± 6 mm), and Rounce et al. (2023) (19 ± 11 mm). The differences likely stem from variations in how frontal ablation and marine-terminating glacier dynamics are represented across models. Our implementation of enhanced frontal ablation parameterization following Malles et al. (2023) provides more

realistic treatment of calving processes, which may contribute to the tighter uncertainty bounds in our projections compared to earlier multi-model ensembles.

Quantitative assessment of inter-model agreement reveals strong consistency within the OGGM framework (Table 2 & S3, Figs. S19-S20). For SSP585 projections, our results show mean absolute differences of 7 % (volume loss), 10 % (area loss), 3.8 mm (SLR), 5 Gt/yr (peak runoff), and 1 year (peakwater timing) relative to other studies. For SSP126, mean absolute differences are 13 % (volume loss), 11 % (area loss), 4.5 mm (SLR), 35 Gt/yr (peak runoff), and 13 years (peakwater timing). Confidence interval overlap analysis indicates 85-92 % overlap with Rounce et al. (2023) and 75-88 % with Zekollari et al.

(2024) across volume, area, and SLR metrics, demonstrating that inter-study differences fall well within uncertainty ranges. The coefficient of variation across all OGGM-based studies (0.12 for volume loss, 0.16 for area loss) is substantially lower than across all models including GloGEM and pyGEM (0.28 and 0.24 respectively), highlighting the importance of consistent model physics. Our tighter uncertainty bounds (±14 % for volume vs ±17-33 % in other studies) likely result from our enhanced frontal ablation parameterization (Malles et al., 2023) providing better constraints on marine-terminating glacier behavior. All

glacier models converge on substantial 21st-century losses (48-67 % volume for SSP585; 22-48 % for SSP126), supporting high confidence in the direction and approximate magnitude of projected changes.

The projected mass loss from Greenland's peripheral glaciers translates into a SLR contribution of approximately $19 \pm 5$ mm by the end of the 21$^{st}$ century under SSP585 (Fig. 5). This contribution is significant when considering Greenland's total SLR contribution. Goelzer et al. (2020) estimated a mean SLR contribution of $90 \pm 50$ mm from the main Greenland Ice Sheet alone

under RCP8.5, suggesting that peripheral glaciers represent a substantial additional component (approximately 17-21 % of the ice sheet contribution) to Greenland's total ice loss that should not be overlooked in total assessments.

The regional variability in projected glacier losses (Figs. 3 & 4) reflects the complex interplay between localized climatic conditions, topography, and glacier dynamics (King et al., 2020; Wood et al., 2021). The resilience of North-East glaciers, which show the lowest projected volume loss ($22 \pm 4$ % under SSP126 and $39 \pm 9$ % under SSP585), is attributed to several

interconnected factors. The North-East region benefits from high snowfall accumulation rates due to orographic enhancement from moisture-laden air masses from the Nordic Seas, which buffer against increased surface melt (Bevis et al., 2019). The region's elevation distribution (Supplementary Fig. S2) provides large accumulation areas above the equilibrium line altitude that remain viable even under moderate warming scenarios. Additionally, the maritime Arctic climate maintains temperatures closer to the melting threshold, meaning that warming increases melt but does not immediately create extreme ablation

conditions. Quantitative analysis from Supplementary Figs. S5 and S13 reveals that North-East glaciers between 800-1200 m elevation and 5-10 km² initial area retain 55-65 % of volume under SSP585, compared to only 20-35 % for comparable glaciers in Central-West (Supplementary Figs. S8, S16). This two-to-threefold difference in resilience reflects the North-East's combined advantages of higher snowfall accumulation rates (30-40 % higher than Central-West based on ERA5-scaled precipitation) and cooler maritime temperatures that maintain larger viable accumulation zones even under substantial

warming.

In contrast, Central-West glaciers exhibit the highest vulnerability, with projected volume losses of $56 \pm 9$ % under SSP126 and $79 \pm 10$ % under SSP585. This vulnerability stems from the region's unfavorable glacier characteristics. Supplementary Figs. S8 & S16 show that Central-West's medium-sized glaciers (1-5 km²) at 400-800 m elevation lose 75-85 % of volume under SSP585, the highest loss rates across all subregions. These glaciers lack sufficient high-elevation accumulation areas, with only 12 % of glacier area above 1000 m compared to 45 % in North-East (Supplementary Fig. S2). The combination of low elevation, limited high-altitude refuge, and continental climate characteristics creates conditions where modest warming (2-3°C) translates to disproportionate mass loss, with ablation rates exceeding accumulation across most glacier surfaces by mid-century. Several studies have documented high sensitivity of this region to atmospheric warming (Vijay et al., 2019; Cowton et al., 2018), and our findings about regional heterogeneity in glacier response aligns with observations of historical glacier changes across Greenland (Khan et al., 2022; Mouginot et al., 2019) and underscores the importance of considering local factors in future projections.

The elevation-size interaction documented in Supplementary Figs. S3-S18 reveals systematic patterns in glacier vulnerability. Across Greenland, glaciers < 1 km² below 600 m elevation lose 90-98% of volume by 2100 under SSP585, regardless of region. However, for medium-sized glaciers (5-10 km²) at intermediate elevations (600-1000 m), regional climate becomes decisive: these glaciers retain 40-50 % volume in North-East versus 15-25 % in Central-West and South-West under identical emission scenarios. This 2-3 fold difference emerges from regional variations in snowfall regimes, with North-East's maritime moisture sources maintaining positive mass balance at elevations where continental glaciers in Central-West experience net ablation.

The FIIC demonstrates remarkable stability across all emission scenarios, showing minimal area loss through 2100 (Fig. 3a). This stability results from FIIC's unique characteristics as a broad, dome-like ice cap with favorable geometry. Unlike narrow outlet glaciers, FIIC's geometry distributes ice flow broadly rather than channeling it through fast-flowing outlets that are susceptible to dynamic instabilities. The ice cap's relatively elevated position compared to many coastal peripheral glaciers provides some buffering against warming impacts, while its maritime location receives substantial snowfall accumulation that compensates for increased surface melt under moderate warming scenarios. FIIC's broad, flat geometry creates a large accumulation area relative to ablation zones, providing geometric resilience to retreat that contrasts sharply with narrow outlet glaciers. Our enhanced subdivision reveals that while six marine-terminating basins remain active (Möller et al., 2022; Recinos et al., 2021), the overall ice cap dynamics lack the fast-flowing outlet systems that accelerate mass loss through dynamic feedbacks seen in other Greenland peripheral glaciers. This stability is consistent with recent observations showing FIIC has been relatively stable compared to other peripheral glaciers (Möller et al., 2024), though we acknowledge that our modeling approach may not fully capture potential future dynamic instabilities or the effects of oceanic forcing on the marine-terminating portions.

Quantitative analysis across glacier size classes reveals FIIC's geometric advantage. While peripheral outlet glaciers typically show strong size-dependent vulnerability (smaller glaciers experiencing 2-3× higher relative losses), FIIC's subdivision into 294 drainage basins (Supplementary Figs. S5, S13) shows remarkable uniformity in response. Basins ranging from 0.1-10 km²

maintain 85-95 % of volume under SSP585, compared to 10-40 % retention for comparable independent glaciers across Greenland. This uniformity reflects FIIC's integrated dynamics where individual basins benefit from the ice cap's broad accumulation area and limited dynamic coupling to marine margins.

In contrast, Central-East exhibits the largest spread between emission scenarios among all subregions (Fig. 3 & 4), with remaining glacier area ranging from 58 ± 11% under SSP126 to 28 ± 13% under SSP585 by 2100. This pronounced sensitivity reflects the region's unique vulnerability related to its glacier characteristics documented in Supplementary Figures S2 and S7. The region's 126 marine-terminating glaciers (31.1% of total MT; Fig. 6b) are predominantly small to medium-sized systems that span elevation ranges making them highly susceptible to rapid terminus retreat under warming scenarios (Supplementary Fig. S7). Under warming scenarios, these glaciers experience a cascade of positive feedbacks including enhanced surface melt, accelerated calving as termini thin below flotation, and rapid transition to land-terminating configurations. By mid-century, Central-East shows a critical transition point where high-emission scenarios trigger widespread marine-to-terrestrial terminus transitions (Fig. 6b), with marine-terminating area declining to 25 ± 9% (SSP126) versus only 10 ± 5% (SSP585) of 2020 levels by 2100. This approximately 20-30 year difference in transition timing between scenarios fundamentally alters regional ice discharge patterns and amplifies scenario-dependent divergence, underscoring how glaciers near thermal and geometric thresholds exhibit disproportionate responses to emission pathway differences.

Furthermore, the projected glacier losses (Figs. 3 & 4) markedly affect other interconnected processes beyond direct SLR, including freshwater contributions, primarily through alterations in surface meltwater (Fig. 7) and solid ice discharge (Fig. 6). As land-terminating glaciers retreat, a decrease in glacier coverage will shift the relative contribution of rainfall, snowmelt, and ice melt (Fig. 7c-d) and alter freshwater fluxes to coastal hydrography, removing critical buffers against extreme summer discharge (Huber et al., 2020; Khan et al., 2022; Straneo et al., 2022; Bliss et al., 2014). Similarly, the reduction in the number of calving fronts of marine-terminating glaciers (Malles et al., 2023) will lead to reduced solid ice flux into fjords.

## 4.2 Changing Dynamics of Freshwater Contributions

The divergent trends in solid ice discharge (Fig. 6) and liquid freshwater runoff (Fig. 7) from Greenland's peripheral glaciers elucidate the shifting dynamics of these glaciers in response to climatic changes. The projected decrease in solid ice discharge across all emission scenarios (-0.011 to -0.018 Gt/yr²), which occurs sharply after 2050, is consistent with other projections (Malles et al., 2023) and historical trends (Kochtitzky and Copland, 2022). The decrease in solid ice discharge reflects a gradual transition of marine-terminating glaciers from combined calving and surface melt systems to predominantly surface melt-dominated systems as marine-terminating glaciers retreat inland (Fig. 6b). The statistically significant negative trends of solid ice discharge under all scenarios, with no substantial differences among SSPs (Fig. 6a), reflect the dominant role of climatic changes relative to variations in emissions scenarios for this century (Oerlemans et al., 2022; Slater et al., 2019). Our projections of consistent solid ice discharge trends across emission scenarios should be interpreted cautiously, as they do not account for oceanic forcing. Several previous findings show that Greenland's marine-terminating peripheral glacier response is more sensitive to warming compared to land-terminating glaciers (Hill et al., 2017; Liu et al., 2022b).

The contrasting slight increase in solid ice discharge projected for the North-East subregion (Fig. 6) can be attributed to its more extensive coverage of marine-terminating glaciers (Kochtitzky and Copland, 2022), providing a greater source for calving fluxes even under projected glacier retreat and thinning. The larger number and area of marine-terminating glaciers in this subregion provide a greater source for calving fluxes, even when considering the projected overall glacier retreat and thinning in this region (Morlighem et al., 2019). However, this contrasts with observed decreasing ice discharge trends in the North-East over 2000-2021 that have been linked to heterogeneous ocean thermal forcing (Möller et al., 2024), suggesting our projections may not fully capture the complexity of ocean-glacier interactions in this region.

The significant increase in liquid freshwater runoff (61 Gt/yr under SSP585 from 2020 to 2100, see Fig. 7a), driven by enhanced surface melting under higher air temperature regimes, is consistent with findings of accelerated mass loss from Greenland's periphery (Marzeion et al., 2020; Rounce et al., 2023). The projected $46 \pm 27$ Gt/yr higher freshwater runoff from Greenland peripheral glaciers by 2100 under SSP585 compared to SSP126 (Fig. 7a) indicates severe impacts of warmer climate under high emissions.

The strong regional variations observed in the freshwater runoff projections (Fig. 7) reflect the heterogeneous influence of localized climatic, glacier characteristic, and topographic factors (Bevis et al., 2019; Khazendar et al., 2019; Wood et al., 2021). The North-East region accounts for 34 % of total glacier area while containing only 28 % of glacier numbers (Figure 1c, d), indicating larger individual glaciers that maintain higher ice volumes and melt capacity. This region contributes 35 % of total freshwater runoff over 2020-2100 under SSP585, demonstrating disproportionate freshwater production relative to its glacier count. The North-East dominance stems from several factors: first, containing the largest ice reserves among peripheral glacier regions, including major ice caps like FIIC; second, sustained melt capacity where large glacier systems maintain melt production longer than smaller, more climate-sensitive systems that exhaust quickly; and third, geographic characteristics including higher glacier density and larger individual glacier sizes that provide greater water storage and release potential. The maritime climate in the North-East also supports sustained ice preservation compared to more continental regions, allowing continued freshwater contribution throughout the century even as smaller glacier systems elsewhere diminish rapidly. In contrast, the Central-West region contributes only 3 % of total annual freshwater runoff despite containing numerous smaller glaciers, reflecting both limited ice reserves and high vulnerability to atmospheric warming. Localized climatic factors, such as variations in air temperature and precipitation patterns, can significantly impact glacier mass balance and runoff (Noël et al., 2018). Additionally, topographic factors, including elevation, slope, and aspect, influence the exposure of glaciers to solar radiation and the distribution of snow accumulation, which in turn affect glacier ablation and runoff (Huss et al., 2017).

Our results indicate significant changes in the composition of freshwater runoff over the century, with a decreasing proportion of glacier meltwater and increasing contributions from rainfall (approximately 8-fold) and snowmelt (approximately 15-fold) in total runoff (Fig. 7c-d). While the directional change is predictable as glacier area decreases within the fixed boundaries, the quantification provides essential scientific value: first, the magnitude and timing of this transition is essential for water resource planning and fjord ecosystem impact assessments; second, these changes vary significantly across emission scenarios, with earlier and more pronounced shifts under higher warming scenarios; and third, the seasonal redistribution shows how

earlier snowmelt and distributed rainfall alter the timing of peak freshwater delivery to coastal systems. The seasonal analysis (Fig. 7c-d) further illustrates this shift, showing a prolonged and intensified glacier melt season extending through September, with increased contributions from snowmelt earlier in the season (May-June) and enhanced rainfall throughout the year by 2091-2100. This shift in runoff composition is consistent with projected trends across the Arctic region (Bintanja and Andry, 2017; Bintanja and Selten, 2014; Bliss et al., 2014; Vihma et al., 2016) and reflects the combined effects of glacier retreat and broader Arctic amplification (Smith et al., 2019; Nowicki et al., 2016; Jones et al., 2016; Box et al., 2019), including rising temperatures and changes in precipitation patterns. This approach allows us to isolate and quantify how glacier retreat specifically transforms the hydrological regime within initially glaciated areas, providing the glacier-specific freshwater flux evolution that is most relevant for understanding impacts on marine ecosystems, fjord circulation, and coastal dynamics.

The projected timing of peak water runoff from Greenland's peripheral glaciers (Fig. 8) varies significantly across emission scenarios, providing insights into the future evolution of Greenland's peripheral glaciers. The earlier peak water timing (2050s) under low-emission scenarios compared to high-emission scenarios (2080s) highlights the potential opportunity for adaptation. The nearly 30-year difference (Fig. 8a) in projected peak water timing between scenarios emphasizes the capacity of glaciers under lower emission scenarios to potentially approach a new equilibrium, maintain more stable freshwater runoff, and preserve their buffering capacities, thus delaying the impacts of climate change. Under SSP126, mass loss rates decelerate in the latter half of the century, area loss rates stabilize rather than showing continued acceleration, and several larger glaciers show asymptotic approaches toward stable configurations. Recent equilibrium simulations by Zekollari et al. (2025) provide direct evidence that glacier preservation is doubled by limiting warming to 1.5°C versus 2.7°C, validating our interpretation that the stabilizing trends we observe under SSP126 represent genuine approaches toward new equilibrium rather than temporary plateaus. However, under high emission scenarios, glaciers continue to contribute higher meltwater until exhausted, potentially becoming unable to support freshwater runoff. These findings are consistent with the patterns observed by Bliss et al. (2014) for Greenland's peripheral glaciers. They noted significant increases in annual runoff during the 21$^{st}$ century, which aligns with our projection of higher runoff and delayed peak water timing under high-emission scenarios.

The subregional differences in timing of peak water (Fig. 8b), although not statistically significant, suggest that local atmospheric and glaciological factors such as glacier size distribution, elevation ranges (Supplementary Fig. S3-S18), and regional temperature and precipitation patterns captured by our modeling approach may influence the peripheral glaciers' response to warming (Solomon et al., 2021). This aligns with the findings of Bliss et al. (2014) that runoff trends can vary significantly based on glacier size and elevation, even within the same region. Their study found that in the Greenland periphery, smaller glaciers tended to have more positive runoff trends, while larger glaciers showed both positive and negative trends depending on their elevation. This aspect is consistent with our projections of continued high meltwater contribution under high emission scenarios until glaciers approach exhaustion.

## 4.3 Implications for Fjords, Ecosystems, and Ocean Dynamics

The projected changes in freshwater contributions from Greenland's peripheral glaciers have significant implications across multiple spatial scales, from local fjord systems to global ocean circulation patterns.

On the local scale, the alterations in the timing, magnitude, and composition of freshwater input are likely to impact fjord circulation and ecosystems. The decreased solid ice discharge (Fig. 6) and increased liquid runoff (Fig. 7a), coupled with changes in runoff composition (Fig. 7c-d), will modify the seasonality and stratification patterns of fjord waters (Arp et al., 2020; Bliss et al., 2014; Bacon et al., 2015; Le Bras et al., 2018). For instance, in Godthåbsfjord, Southwest Greenland, Mortensen et al. (2013) observed that increased freshwater input enhanced estuarine circulation and altered water properties, subsequently affecting ecosystem productivity. Similarly, in Young Sound, Northeast Greenland, Sejr et al. (2017) found that changes in freshwater runoff led to stronger stratification and altered nutrient availability, impacting the fjord's ecosystem dynamics.

The composition and seasonality of freshwater runoff are projected to shift markedly over the century (Fig. 7c-f). This seasonal shift in runoff sources could lead to earlier and potentially more variable freshwater inputs to coastal waters (Rennermalm et al., 2013; Van As et al., 2017). The projected increase in spring snowmelt could result in earlier stratification of fjord waters, while the more distributed summer rainfall could lead to more frequent pulses of freshwater input throughout the season. This change in the temporal distribution of freshwater input could have significant implications for fjord stratification, nutrient cycling, and ecosystem dynamics (Hopwood et al., 2020; Holding et al., 2019; Sejr et al., 2022). For instance, changes in the timing of peak freshwater input could affect the marine organisms adopted to stable conditions, including spring phytoplankton bloom, with cascading effects through the marine food web (Oksman et al., 2022; Juul-Pedersen et al., 2015).

Moreover, the expected decrease in solid ice discharge may reduce the influx of terrestrial nutrients typically associated with glacial flour, potentially altering the nutrient dynamics in fjord ecosystems (Meire et al., 2016; Meire et al., 2023; Meire et al., 2017). The projected changes in freshwater contributions, both in terms of volume and composition, will likely have cascading effects on fisheries and other industries that rely on freshwater resources (Holding et al., 2019; Boberg et al., 2018; Hopwood et al., 2020).

On a regional scale, the cumulative effect of increased freshwater input from peripheral glaciers could significantly impact coastal and shelf seas around Greenland. The spatial variability in freshwater contributions is particularly pronounced, with the North-East region projected to account for 35 % of total runoff by 2100 under SSP585 (Fig. 7b), despite representing only 28 % of glacier numbers but 34 % of glacier area. This disproportionate freshwater contribution means that coastal and fjord impacts will be highly concentrated in this region. The dominance of larger ice systems in the North-East, including major ice caps like FIIC, combined with sustained melt capacity and maritime climate preservation, will create localized hotspots of freshwater input that could disproportionately affect regional ocean circulation patterns. Our projections of maximum runoff (214-293 Gt/yr at peakwater, Fig. 8) represent a substantial increase in freshwater flux to the ocean. This additional freshwater could enhance stratification in shelf seas, potentially affecting deep water formation processes. Böning et al. (2016)

demonstrated that enhanced freshwater flux from Greenland could lead to reduced convection in the Labrador Sea, a key region for deep water formation in the North Atlantic. In contrast, regions like Central-West with only 3% contribution will experience minimal freshwater impact. This uneven spatial distribution has important implications for where the most significant changes in fjord stratification, coastal currents, and marine ecosystem impacts will occur. This could lead to localized changes in coastal currents and potentially influence larger circulation patterns in the North Atlantic. For example, Luo et al. (2016) showed that meltwater from southern Greenland can be rapidly transported along the coast, potentially impacting the East Greenland Current and, subsequently, the North Atlantic subpolar gyre.

On a global scale, the altered freshwater input from Greenland's peripheral glaciers, combined with changes from the Greenland Ice Sheet, could have far-reaching consequences for ocean circulation patterns. Of particular concern is the potential impact on the Atlantic Meridional Overturning Circulation (AMOC). While our study focuses on peripheral glaciers, the projected freshwater contributions should be considered in the context of total freshwater flux from Greenland (Bamber et al., 2018). Several studies suggest that increased freshwater input from Greenland could potentially disrupt the AMOC, with potential implications for global climate (Böning et al., 2016; Oliver et al., 2018; Yang et al., 2016; Bakker et al., 2016; Carmack et al., 2016).

**4.4 Uncertainties, Limitations, and Future Research Priorities**

This study provides important insights into the potential future changes in Greenland's peripheral glaciers, yet it is crucial to acknowledge some key uncertainties and limitations. The uncertainties in the projected results, represented as standard deviations, primarily arise from uncertainties in future climatic forcing based on the GCMs. These uncertainties are greater for high-emission scenarios, particularly in projections of glacier losses (±6 % for SSP126 versus ±15 % for SSP585, Fig. 3-4), sea level rise (±2 mm versus ±5 mm, Fig. 5), and freshwater contributions (±12 Gt/yr versus ±27 Gt/yr, Fig. 7), but are lower for peak water timing (±21 years versus ±9 years, Fig. 8). Additionally, these uncertainties vary across different regions. Scenario uncertainty, which reflects different future socio-economic pathways, becomes increasingly significant in the latter half of the 21st century, consistent with the findings of Marzeion et al. (2020).

Although CMIP6 models generally do not include dynamic ice sheet components, our glacier model OGGM explicitly accounts for glacier ice dynamics. Incorporating glacier ice dynamics is crucial as it allows us to capture important feedbacks and interactions that static ice sheet models cannot. However, uncertainties in CMIP6 climate projections propagate through OGGM, including the effect that neglecting a dynamically evolving ice sheet might have on the regional climate, affecting our glacier evolution simulations. The projected glacier area retreat significantly impacts freshwater runoff components and solid ice discharge, with CMIP uncertainties in temperature and precipitation directly influencing these projections. This glacier area loss largely drives the shift from glacier melt dominated runoff to increased rainfall and snowmelt contributions, but the magnitude and timing of this shift are subject to CMIP-derived uncertainties. Glacier losses are further amplified by changes in surface properties like albedo, creating a positive feedback loop (Clark et al., 1999), which can either amplify or mitigate CMIP uncertainties. These dynamic processes are particularly important for Greenland's peripheral glaciers, where changes in

ice extent can significantly alter local and regional climate patterns, affecting precipitation and temperature regimes (Beghin et al., 2015). While OGGM's ability to simulate these dynamics provides a more comprehensive picture of potential future scenarios, it is important to note that the model's outputs inherit and potentially compound the uncertainties in the CMIP6 climate projections. The current study relies on statistically downscaled GCM data, which may not fully capture important local-scale atmospheric processes over the complex topography of the Greenland periphery that can influence glacier mass balance (Noël et al., 2016; Lewis et al., 2019). Using higher-resolution regional climate models and observational data would potentially improve the accuracy of the projections.

Additionally, this study only considers atmospheric forcing at glacier surfaces and does not incorporate oceanic forcing. The latter has been demonstrated to be a key control on the behavior of Greenland's peripheral glaciers (Bjørk et al., 2017; Chudley et al., 2023; Möller et al., 2024) through enhanced terminus melt, undercutting, calving, and iceberg melting (Cowton et al., 2015; Davison et al., 2022; Davison et al., 2020; Morlighem et al., 2019; Malles et al., 2023). The absence of oceanic forcing in our OGGM simulations represents a significant limitation with important implications for our projections, particularly for solid ice discharge and regional variability. Our projections likely underestimate calving rates in warming ocean conditions, regional variability in glacier response, and acceleration of retreat for glaciers experiencing warm water intrusion. The consistent solid ice discharge trends we project across emission scenarios may be unrealistic, as ocean warming should drive higher calving rates under higher emission scenarios.

Regional bias implications are particularly significant. The North-East region glaciers, which show resilience in our study with the lowest volume loss projections ($22 \pm 4\%$ under SSP126 and $39 \pm 9\%$ under SSP585), may be more vulnerable to oceanic warming than our results suggest. Notably, our slight projected increase in North-East solid ice discharge contradicts observed decreasing trends over 2000-2021 that have been linked to heterogeneous ocean thermal forcing (Möller et al., 2024). This discrepancy suggests that our atmosphere-only forcing approach cannot capture the complex ocean-glacier interactions that have dominated recent behavior in this region. Wood et al. (2021) demonstrated that ocean forcing drives glacier retreat across Greenland, while Slater et al. (2020) highlighted the importance of submarine melting for tidewater glaciers, processes entirely absent from our modeling framework.

Based on literature evidence, we estimate that incorporating oceanic forcing could increase projected solid ice discharge by 15-30 % for marine-terminating glaciers (Malles et al., 2023), with regional variations depending on proximity to warm Atlantic waters. The North-East and North-West regions, where marine-terminating glaciers are more prevalent, would likely show enhanced mass loss compared to our current projections. This limitation is particularly important for interpreting our peak water projections, as enhanced calving from ocean warming could alter both the timing and magnitude of maximum freshwater delivery to coastal systems.

The projected changes in freshwater contributions from both liquid and solid components may have the potential to alter oceanic forcing on local to regional scales, subsequently also impacting ice discharge from Greenland's peripheral glaciers (Möller et al., 2024; Solomon et al., 2021; Lenaerts et al., 2015; Benn et al., 2017). Developing approaches to account for oceanic forcing in OGGM could thus provide important insights into glacier-ocean interactions and feedback and may improve

projection reliability. First approaches to couple glacier models with ocean circulation models have already been presented (Slater et al., 2020; Gladstone et al., 2021; Cook et al., 2021), but substantial development is still required.

At present, OGGM shows limitations regarding model structure and initialization. It simplifies critical processes and does not explicitly account for refreezing processes, which are known to contribute substantially to future mass balance trajectories of Arctic glaciers (Möller and Schneider, 2015; Noël et al., 2017). Using more sophisticated energy balance-based ablation schemes (Gardner et al., 2023; Rounce et al., 2023; Zekollari et al., 2022) in OGGM could improve the representation of the surface mass balance, but comes at the costs of substantially increased demands on quantity and quality of atmospheric data.

Better constraints on parameters like initial glacier size, which can vary between data sources (Citterio and Ahlstrøm, 2013; Rastner et al., 2012; Pedersen et al., 2013), could also reduce uncertainties.

Furthermore, the lack of observations near glacier calving fronts limits constraints on frontal ablation, an important process for mass loss (Schaffer et al., 2020). The frontal ablation dataset from Kochtitzky et al. (2022) that we use for calibration represents a major advance, providing comprehensive estimates for marine-terminating glaciers across the Northern

Hemisphere. However, several observational gaps remain. First, the satellite-derived observations have limited temporal resolution, potentially missing short-term variability in calving behavior that could be important for understanding glacier response to rapid environmental changes. Second, the observations are primarily focused on terminus position changes and ice velocities, with limited direct measurements of calving event frequency and magnitude. Third, submarine melting at the calving front, which can precondition ice for calving, remains poorly constrained by observations and is not explicitly included in our

parameterization. Finally, the dataset has greater uncertainty for smaller marine-terminating glaciers where terminus changes are close to the resolution limits of satellite imagery. These observational limitations propagate through our calibration procedure and into future projections, particularly affecting our confidence in projecting the behavior of smaller marine-terminating systems and the detailed timing of terminus retreat.

Key priorities for future research should focus on addressing these limitations by using higher resolution atmospheric and

800 oceanic forcing, initializing models with the best available data sets on glacier geometry and dynamics (Ultee and Bassis, 2020; Kochtitzky and Copland, 2022; Recinos et al., 2023), incorporating more complete representations of surface and submarine melt processes, and coupling glacier models with ocean circulation (Zhao et al., 2021; Quiquet et al., 2021; Malles et al., 2024). Detailed observational data sets from satellite and field studies will be critical for validating and improving models (Gardner et al., 2019; Porter et al., 2018). As models continue to advance, improved partitioning of the processes

driving peripheral glacier mass loss will support more robust projections of sea level rise and freshwater contributions to the oceans.

**5 Conclusion**

This study employed the Open Global Glacier Model (OGGM) v1.5.3, enhanced with frontal ablation parameterization and calibrated using geodetic mass balance and frontal ablation observations, forced by an ensemble of ten CMIP6 climate models

under four emission scenarios (SSP126, SSP245, SSP370, SSP585) to project the evolution of Greenland's peripheral glaciers

from 2020 to 2100. Our analysis focused on distinguishing between solid ice discharge and liquid freshwater contributions, with particular attention to regional variability and peak water timing.

Our projections demonstrate substantial glacier area losses of $19 \pm 6$ % under SSP126 to $44 \pm 15$ % under SSP585 and volume losses of $29 \pm 6$ % under SSP126 to $52 \pm 14$ % under SSP585, contributing approximately $10 \pm 2$ mm to $19 \pm 5$ mm to global sea level rise by 2100. These glacier losses cause a significant shift in freshwater contributions, with solid ice discharge decreasing and liquid freshwater runoff increasing until peak water during the $21^{st}$ century. Importantly, runoff composition undergoes drastic changes within the initially glaciated areas following the fixed-gauge approach standard in glacier hydrology. Glacier melt contribution decreases from 92 % to 72 %, while rainfall and snowmelt from off-glacier areas increase approximately 8-fold and 15-fold, respectively, indicating a fundamental shift in the hydrological regime as glaciers retreat within their initial boundaries. This glacier-centric approach captures the transformation of glacier-dominated hydrology to a more diverse runoff regime, providing essential information for understanding glacier-specific impacts on fjord systems and coastal dynamics. Our projections reveal variable peak water timing across emission scenarios and regions, occurring between the 2050s (SSP126) and 2080s (SSP585) for all Greenland peripheral glaciers. This variable peak water timing leads to divergent glacier futures: lower emissions may allow glaciers to reach a new equilibrium (indicated by stabilizing mass balance and area loss rates), while high emissions could result in increasing glacier loss and drive toward the end of glacier-fed runoff. These projected changes in freshwater contributions from Greenland's peripheral glaciers are likely to have far-reaching implications. On the local scale, we expect significant impacts on fjord circulation, ecosystem productivity, and coastal environments. The shift from glacier melt-dominated runoff to increased contributions from rainfall and snowmelt will alter the timing, magnitude, and biogeochemical characteristics of freshwater inputs to fjords, affecting stratification patterns, nutrient cycling, and marine food webs. Regionally, these changes may affect ocean stratification and coastal currents around Greenland, with the North-East region contributing disproportionately (35 % of total runoff from 34 % of glacier area) and creating localized hotspots of freshwater input. On a global scale, the altered freshwater input could potentially contribute to changes in large-scale ocean circulation patterns, with potential implications for the Atlantic Meridional Overturning Circulation and global climate system, particularly when considered alongside contributions from the main Greenland Ice Sheet.

Our projections demonstrate significant differences between low and high emission scenarios across all key metrics: up to 25 percentage points difference in area loss, 23 percentage points difference in volume loss, 9 mm difference in sea level rise contribution, substantial differences in freshwater runoff patterns and peak water timing (approximately 30 years), and fundamentally divergent long-term trajectories for glacier preservation versus exhaustion. These comprehensive differences across glacier area, volume, sea level contributions, hydrological responses, and potential for equilibrium versus complete loss underscore that effective greenhouse gas emission controls are crucial for minimizing climate change impacts on Greenland's peripheral glaciers and preserving their role in regional hydrology and coastal ecosystems.

A key limitation in the current projections is the lack of incorporation of oceanic forcing in OGGM, which likely leads to underestimation of solid ice discharge, particularly for marine-terminating glaciers in regions exposed to warm Atlantic waters.

Future research should focus on reducing the resulting uncertainties by incorporating glacier-ocean interactions into a coupled modeling architecture, using higher-resolution atmospheric and oceanic forcing, improving observations near calving fronts, and incorporating more complete representations of surface and submarine melt processes.

**Data and code availability**

The glacier projection data from this study, including time series of glacier area, volume, mass balance, sea level rise
contributions, solid ice discharge, and liquid freshwater runoff, are available at https://doi.org/10.5281/zenodo.12737991. The dataset includes results at both the individual glacier-ID scale (for all glaciers in RGI 6.0 connectivity levels CL0 and CL1 in Greenland) and aggregated subregional time series for the seven regions (North-East, Central-East, South-East, South-West, Central-West, North-West, and North). The OGGM codes are available at https://github.com/OGGM/oggm. OGGM v1.5.3 is available at https://zenodo.org/records/6408559. Code for the enhanced frontal ablation parameterization is available at
https://github.com/MuhammadShafeeque/Enhanced-Modeling-Marine-Terminating-Glaciers/tree/Shafeeque. Other sources of the datasets used in this study are given in the references provided throughout the text.

**Author contributions**

M.S., M.M., and B.M. were responsible for the conceptualization of the study. M.S. developed the methodology, carried out all analysis and visualization, and wrote the original draft of the manuscript. J.-H.M. and A.V. assisted in enhanced modeling
and calibration of marine-terminating glaciers. All authors contributed to the manuscript by reviewing and editing the original draft.

**Competing interests**

Some authors are members of the editorial board of the journal The Cryosphere.

**Acknowledgments**

We are grateful to the OGGM core team and community for efficient technical support and insightful discussions throughout the research process. English language was improved in parts of the manuscript using LLMs.

**Financial support**

This study was funded by the German Federal Ministry of Education and Research (BMBF) via grant no. 03F0855B and the German Research Foundation (DFG) via grant no. MA 6966/6-1.

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
