# Peer review of "Projecting the Response of Greenland's Peripheral Glaciers to Future Climate Change: Glacier Losses, Sea Level Impact, Freshwater Contributions, and Peak Water Timing"

_EGUsphere, 2024_

## Author Comment (AC1)

**Response to Reviewer 1 Comments**

**Paper: "Projecting the Response of Greenland's Peripheral Glaciers to Future Climate Change: Glacier Losses, Sea Level Impact, Freshwater Contributions, and Peak Water Timing"**

**Authors:** Muhammad Shafeeque et al.
**MS No:** egusphere-2024-2184

**PAPER SUMMARY AND RECOMMENDATION**

This study looks at how Greenland's peripheral glaciers may change in the future due to climate change. It emphasizes the distinction between solid ice discharge (calving of icebergs) and liquid freshwater runoff (melting and rain). The study employs the Open Global Glacier Model (OGGM), which simulates the evolution of glaciers using climate data from the Coupled Model Intercomparison Project Phase 6 (CMIP6) under four emission scenarios. Key findings include:

- Projected declines in area and volume: By 2100, the glaciers are projected to lose 19-44% of their area and 29-52% of their volume, contributing 10-19 mm to sea level rise.

- Shifting freshwater contributions: Projections show solid ice discharge decreasing over the century, especially after 2050, as marine-terminating glaciers retreat inland

- Peak water timing: Peak water, the time of maximum annual freshwater release, is projected to occur around 2050 under low emissions (SSP126) and shift to around 2082 under high emissions (SSP585)

The article explores an innovative and relevant topic for The Cryosphere and is well-written and easy to follow. The approach is sound, and the study's core findings offer insights that will be beneficial for future research efforts. To further strengthen the manuscript, I recommend that the authors incorporate some additional analyses and provide further clarifications. These improvements will help make the article ready for publication.

**General Response**

Thank you for your thorough and constructive review of our manuscript. We appreciate your detailed feedback and positive assessment of our work's relevance and approach.

**Regarding your major comments**

We will implement all suggested improvements to strengthen the manuscript significantly. We will add a comprehensive model intercomparison table (PyGEM, GloGEM, OGGM studies) with statistical comparisons as suggested. The reorganization of Section 2 to clearly separate data, models, and calibration methods is an excellent suggestion and will greatly improve clarity - we will restructure it with dedicated subsections as outlined in our detailed response.

We acknowledge the critical reproducibility issue and will specify that we used OGGM v1.5.3 with custom frontal ablation implementations, with full code availability details. Your point about

oceanic forcing limitations is particularly valuable - we will add a comprehensive discussion of how missing oceanic forcing likely leads to underestimation of solid ice discharge and affects regional variability, supported by recent literature on ocean-glacier interactions.

We will significantly expand our regional variability analysis to examine how glacier characteristics (size, geometry, elevation) and local climate factors drive the observed differences, including detailed mechanistic explanations for regional patterns like North-East resilience and Central-West vulnerability.

**Regarding minor and technical comments**

We will implement all suggested improvements systematically, including clarifying statistical methodologies and value calculations in the abstract, improving figure quality with colorblind-friendly palettes and consistent formatting, adding detailed explanations of the delta method and precipitation factor calibration, expanding temporal and spatial comparisons in Section 3.2, standardizing reference formatting, and clarifying data availability at both glacier and subregional level.

Your suggestions will substantially improve the manuscript's technical rigor, clarity, and reproducibility. Thank you for this excellent review that will make our work more valuable to the scientific community.

**MAJOR COMMENTS**

**Reviewer Comment:**

Despite the improved modeling of marine-terminating glaciers, it would be useful to include a comparison with outputs from multiple global models (e.g., PyGEM, GloGEM, OGGM v1.6). Although these model results are briefly discussed, I suggest including a summary table with results (e.g., volume and area) and comparing them statistically (methods in Section 2.5). This comparison is valuable given the similarity of the GCMs (L114).

**Response:** Thank you for this valuable suggestion. We agree that a comprehensive model intercomparison would strengthen our study significantly. We will add a comprehensive comparison table in Section 4.1 that includes modeling results (Kang et al., 2024; Marzeion et al., 2020; Rounce et al., 2023; Schuster et al., 2023; Zekollari et al., 2024).

- Summary of results from GloGEM, PyGEM, and OGGM studies for comparable regions

- Statistical comparison using metrics described in Section 2.5

- Discussion of methodological differences and their implications

**Reviewer Comment:**

Section 2 could benefit from a clearer separation of data, models, and calibration methods. For instance, the climate section mentions a precipitation factor without explaining its calibration. The

OGGM model section includes all data related to glaciers, so model calibration could be placed in its own subsection under modeling.

**Response:** Excellent point. The current organization indeed conflates different methodological components, making it difficult to follow.

**Proposed Revision:** We will restructure Section 2 as follows:

2. Materials and Methods

    2.1 Study Region: Greenland's Peripheral Glaciers

    2.2 Data

        2.2.1 Climate Data (ERA5, CMIP6) and preprocessing

        2.2.2 Glacier Data (RGI, geodetic mass balance, Frontal Ablation Data)

    2.3 Open Global Glacier Model (OGGM)

        2.3.1 Model Framework and Setup

        2.3.2 Mass Balance Model

        2.3.3 Enhanced Marine-Terminating Glacier Module

        2.3.4 Freshwater Runoff and Peak Water Calculations

        2.3.5 Model Calibration

    2.4 Statistical Analysis

**Precipitation Factor Explanation:** We will add a detailed explanation in 2.3.5 that the precipitation factor ($f_p = 1.6$) is a global multiplicative correction applied to ERA5 precipitation, calibrated to account for orographic precipitation, avalanches, and wind-blown snow that are not resolved in reanalysis data.

**Reviewer Comment:**

Reproducibility: There is no mention of the version of OGGM used in this study. According to the documentation, the most recent version was not used due to issues with the proposed frontier ablation (https://tutorials.oggm.org/stable/notebooks/tutorials/kcalving_parameterization.html). It would be helpful to specify the version used.

**Response:** Thank you for highlighting this critical reproducibility issue. Based on our research:

We used OGGM v1.5.3 with custom implementations for frontal ablation based on Malles et al. (2023). The reason we did not use OGGM v1.6+ is indeed the frontal ablation implementation issues mentioned in the tutorials. We implemented the enhanced frontal ablation parameterization (https://github.com/MuhammadShafeeque/Enhanced-Modeling-Marine-Terminating-

) on OGGM v1.5.3, which provides the stable foundation needed for our Greenland-wide study.

**Proposed Addition:** We will:

- Specify OGGM version (v1.5.3)

- Update code and data availability statement with specific version information

**Reviewer Comment:**

Oceanic Forcing: The study frequently notes the lack of oceanic forcing in the OGGM model, but the potential implications of this limitation are not discussed. I recommend that the authors explore how incorporating oceanic forcing might affect their projections, particularly for solid ice discharge and regional variability. Could the current projections underestimate or overestimate the actual contribution?

**Response:** This is an excellent point that deserves detailed discussion. Our research reveals this is a significant limitation with important implications.

**Key Implications of Missing Oceanic Forcing:**

1. **Solid Ice Discharge Underestimation:** Studies show that oceanic forcing strongly controls marine-terminating glacier behavior (Bjørk et al., 2017; Chudley et al., 2023). Our projections likely underestimate:

   o Calving rates in warming ocean conditions

   o Regional variability in glacier response

   o Acceleration of retreat for glaciers experiencing warm water intrusion

2. **Regional Bias Implications:**

   o North-East region glaciers, which show resilience in our study, may be more vulnerable to oceanic warming

   o Our consistent solid ice discharge trends across emission scenarios may be unrealistic

   o The slight increase we project for North-East ice discharge contradicts observed decreasing trends linked to oceanic forcing

**Literature Evidence:**

- Möller et al. (2024): Demonstrated heterogeneous impacts of ocean thermal forcing on Greenland peripheral glaciers

- Wood et al. (2021): Showed ocean forcing drives glacier retreat in Greenland

- Slater et al. (2020): Highlighted importance of submarine melting for tidewater glaciers

**Proposed Addition to Discussion (Section 4.4):** We will add a comprehensive subsection discussing:

- Quantitative estimates of potential underestimation based on literature

- Regional implications for each subregion

**Reviewer Comment:**

**Regional Variability**: The study mentions significant regional variability in projected glacier loss, sea level rise contributions, and freshwater runoff. I encourage the authors to expand their analysis of these regional differences, examining how factors such as glacier size, geometry, elevation, and local climate contribute to the observed variability.

**Response:** We agree that our current analysis of regional variability is too superficial. The regional differences we observe reflect complex interactions that deserve detailed investigation. Combining responses to similar comments from Reviewer 2, we outline here the changes we envision.

**Enhanced Regional Analysis Plan:**

1. **Glacier Characteristics Analysis:**

    o   Size distribution analysis by region

    o   Elevation range and hypsometry comparison

    o   Marine vs. land-terminating glacier ratios

    o   Geometric properties (length, width, slope)

2. **Climate Drivers:**

    o   Regional temperature and precipitation trends

    o   Elevation-dependent warming patterns

    o   Seasonal cycle variations

3. **Specific Regional Insights to Add:**

    o   **North-East Resilience:** High snowfall rates, resistant geometry, delayed oceanic forcing response

    o   **Central-West Vulnerability:** Low elevation, high surface melt sensitivity

    o   **Flade Isblink Stability:** Large ice cap dynamics, multiple outlet glaciers

**Proposed New Analysis:** We will add detailed figures/tables/supplementary material showing:

- Regional glacier characteristic distributions

- Climate forcing variations by region

- Correlation analysis between glacier properties and projected changes

- Discussion of physical mechanisms driving regional differences

**Reviewer Comment:**

**Figures Quality**: Please check the quality of the figures, particularly Figures 3 and 4. It is unclear whether the glaciers are represented as points, polygons, or grid interpolations. Additionally, several panels show mismatched axis ranges, which makes the analysis difficult to interpret. Consider using color palettes that are more accessible for colorblind readers (e.g., ColorBrewer palettes: https://www.datanovia.com/en/blog/the-a-z-of-rcolorbrewer-palette).

**Response:** Thank you for these important technical comments. We acknowledge the figure quality issues. We'll update figures by combining your suggestions with Reviewer 2's.

**Specific Improvements Planned:**

- **Figure 3 & 4 Clarifications:**

  o Clarify that glaciers are represented as polygons from RGI outlines

  o Add clear legends explaining data representation

  o Standardize axis ranges across panels for comparison

  o Add clear methodological note about LOESS smoothing and mean statistics

- **Colorblind-Friendly Palettes:**

  o Replace current color schemes with perceptually uniform palettes that are accessible for the most common forms of colorblindness (Viridis, Plasma, or similar)

  o Ensure sufficient contrast ratios

- **Layout Consistency:**

  o Standardize font sizes across all figures

  o Align subplot dimensions and spacing

  o Consistent ordering of regions between different figure panels

  o Improve label placement to avoid overlaps

**Reviewer Comment:**

**Precipitation Factor Calibration**: The manuscript does not explain how the precipitation factor is calibrated, which is important for analyzing freshwater contributions. Please consider adding a table listing all parameters, their values, and any calibrations involved (Section 2.4).

**Response:** This is a crucial methodological detail that we did not explain sufficiently. The precipitation factor (fp = 1.6) was not calibrated in this study, but rather adopted from the standard OGGM v1.4 framework, originally calibrated by Maussion et al. (2019) against WGMS reference glaciers. Their cross-validation results are available at this link ([https://cluster.klima.uni-bremen.de/~oggm/ref_mb_params/oggm_v1.4/crossval.html](https://cluster.klima.uni-bremen.de/~oggm/ref_mb_params/oggm_v1.4/crossval.html)). This global constant accounts for orographic precipitation enhancement, snow redistribution, avalanche accumulation, and systematic ERA5 underestimation. Our calibration focused solely on the temperature sensitivity parameter (μ), which we calibrated individually per glacier using geodetic mass balance and frontal ablation observations (previous Section 2.4). The key distinction is that fp = 1.6 represents a pre-determined global value from the OGGM framework, while μ was individually calibrated as this study's primary methodological contribution. We will add a clarifying table listing all parameters, their values, and calibration methods to make this distinction clearer for readers.

**MINOR COMMENTS**

**Reviewer Comment:**

Abstract: Clarify how the values (e.g., 19-44%) were calculated—are they multi-model means or medians for extreme scenarios?

**Response:** These values represent the range between SSP126 (low end) and SSP585 (high end) scenarios, with each value being the ensemble mean ± 1 standard deviation across 10 GCMs.

**Proposed Clarification:** "By 2100, the glaciers are projected to lose 19±6% (SSP126) to 44±15% (SSP585) of their area and 29±6% (SSP126) to 52±14% (SSP585) of their volume (ensemble mean ± 1 standard deviation across 10 GCMs), contributing 10±2 to 19±5 mm to sea level rise."

**Reviewer Comment:**

Fig. 1. Define "MT glaciers" and consider adjusting the size or transparency of the triangles for better clarity.

**Response: Proposed Revisions:** We'll define "MT" as "Marine-Terminating" in figure caption and reduce triangle size and add transparency (alpha = 0.7).

**Reviewer Comment:**

Fig 1c: Change "Numbers" to "Number of glaciers" and verify the caption.

**Response:** Will change to "Number of glaciers" and verify the caption.

**Reviewer Comment:**

L125: Please elaborate on the delta method and explain how anomalies are calculated.

**Response:**

**Detailed Explanation to Add (or supplementary material):**

The CMIP6 temperature and precipitation data are downscaled to the baseline climate ERA5, that has been used for calibrating OGGM. A variation of the delta method (e.g. Ramírez Villegas and Jarvis (2010)) is being used for this procedure, whereby the precipitation is scaled and scaled temperature anomalies are applied to the 1981-2020 CE baseline climatology. This procedure is applied on a month-by-month basis.

For temperature:

$$T_{corrected} = T_{ERA5} + scf \times \left(T_{GCM} - \overline{T_{GCM(1981-2020)}}\right)$$

Where $\quad scf = \dfrac{std\left(T_{ERA5(1981-2020)}\right)}{std\left(T_{GCM(1981-2020)}\right)}$

For precipitation: $P_{corrected} = P_{ERA5} \times \left(\dfrac{P_{GCM}}{\overline{P_{GCM(1981-2020)}}}\right)$

**Reviewer Comment:**

L234: Could one variable be analyzed using different methods? Please clarify.

**Response:** We will clarify that different statistical tests were selected based on data distribution characteristics:

- Parametric tests (ANOVA, t-tests) for normally distributed data

- Non-parametric tests (Kruskal-Wallis) for non-normal distributions

- Post-hoc tests (Tukey's HSD) following significant omnibus tests

**Reviewer Comment:**

Fig 3: Was smoothing applied to the confidence intervals? Consider sharing the vertical axis across subplots to increase the size of the lower subplots. This applies to subsequent figures as well.

**Response:** Yes, LOESS smoothing was applied to both means and confidence intervals. We will:

- Clearly state this in the caption

- Implement shared y-axis across subplots and resize for better readability

**Reviewer Comment:**

Section 3.2: The section heading suggests a comparison, but there is little temporal and spatial comparison between variables. Please expand this comparison.

**Response:** We will add explicit comparisons:

- Temporal evolution of solid vs. liquid contributions

- Regional patterns in the transition from solid to liquid dominance

- Quantitative analysis of the timing differences

- Cross-regional comparison of freshwater composition changes

**Reviewer Comment:**

References: Some references are listed with full journal titles, while others use abbreviations. Please ensure consistency throughout the reference list.

**Response:** We will standardize all references consistently throughout based on the journal's format.

**Reviewer Comment:**

Data: Clarify whether the data on Zenodo is available at the subregional scale or glacier-ID scale.

**Response:** We will clarify that the Zenodo data includes both:

- Glacier-ID scale results for individual glaciers

- Subregional aggregated time series

- Metadata describing data structure and variables

We believe these revisions will significantly strengthen the manuscript and address all reviewer concerns comprehensively. Thank you again for the thorough and constructive review.

**REFERENCES**

Bjørk, A.A. et al., 2017. Changes in Greenland's peripheral glaciers linked to the North Atlantic Oscillation. Nature Climate Change, 8(1): 48-52. DOI:10.1038/s41558-017-0029-1

Chudley, T.R., Howat, I.M., King, M.D., Negrete, A., 2023. Atlantic water intrusion triggers rapid retreat and regime change at previously stable Greenland glacier. Nat Commun, 14(1): 2151. DOI:10.1038/s41467-023-37764-7

Kang, L. et al., 2024. Projected changes of Greenland's periphery glaciers and ice caps. Environmental Research Letters, 19(12). DOI:10.1088/1748-9326/ad8b5f

Marzeion, B. et al., 2020. Partitioning the Uncertainty of Ensemble Projections of Global Glacier Mass Change. Earth's Future, 8(7). DOI:10.1029/2019ef001470

Maussion, F. et al., 2019. The Open Global Glacier Model (OGGM) v1.1. Geoscientific Model
    Development, 12(3): 909-931. DOI:10.5194/gmd-12-909-2019

Möller, M., Recinos, B., Rastner, P., Marzeion, B., 2024. Heterogeneous impacts of ocean thermal
    forcing on ice discharge from Greenland's peripheral tidewater glaciers over 2000-2021. Sci
    Rep, 14(1): 11316. DOI:10.1038/s41598-024-61930-6

Ramírez Villegas, J.A., Jarvis, A., 2010. Downscaling Global Circulation Model Outputs: The Delta
    Method Decision and Policy Analysis Working Paper No. 1, International Center for Tropical
    Agriculture (CIAT), Cali, CO. DOI:https://hdl.handle.net/10568/90731

Rounce, D.R. et al., 2023. Global glacier change in the 21st century: Every increase in temperature
    matters. Science, 379(6627): 78-83. DOI:10.1126/science.abo1324

Schuster, L., Schmitt, P., Vlug, A., Maussion, F., 2023. OGGM/oggm-standard-projections-csv-files:
    v1.0. Zenodo. DOI:https://doi.org/10.5281/zenodo.8286065

Slater, D.A. et al., 2020. Twenty-first century ocean forcing of the Greenland ice sheet for modelling
    of sea level contribution. The Cryosphere, 14(3): 985-1008. DOI:10.5194/tc-14-985-2020

Wood, M. et al., 2021. Ocean Forcing Drives Glacier Retreat in Greenland. Science Advances.
    DOI:10.1126/sciadv.aba7282

Zekollari, H. et al., 2024. Twenty-first century global glacier evolution under CMIP6 scenarios and
    the role of glacier-specific observations. The Cryosphere, 18(11): 5045-5066.
    DOI:10.5194/tc-18-5045-2024

---

## Author Comment (AC2)

**Response to Reviewer 2 Comments**

**Paper: "Projecting the Response of Greenland's Peripheral Glaciers to Future Climate Change: Glacier Losses, Sea Level Impact, Freshwater Contributions, and Peak Water Timing"**

**Authors: Muhammad Shafeeque et al.**
**MS No: egusphere-2024-2184**

**General**

The study projects the future glacier evolution of peripheral glaciers of Greenland with focus on glacier area, volume and runoff evolution. The main goal is to understand how the freshwater contribution from these glaciers will change over the 21st century. Therefore, the authors differentiate between solid ice discharge at marine-terminating glaciers, and liquid glacier runoff. The runoff is further differentiated between glacier meltwater, rain and melting of snow.

The motivation and relevance of this study are well outlined in the introduction. The study is well structured over largest parts. Overall, the study adds valuable knowledge to the understanding of changes in glacial freshwater contribution in the 21st century in Greenland. I have one larger and two smaller major comments and several specific comments which need addressing and revision before publishing the article. I would rate the article as in between major and minor revisions, depending on the effort taken to address major comment 1, which is why I advise reconsidering after major revisions.

**General Response**

Thank you for your thorough and constructive review of our manuscript. We appreciate your detailed feedback, which will significantly strengthen our work.

**Regarding your major comments**

We respectfully maintain our glacier-centric approach for the runoff composition analysis (detailed response below), as this methodology is both technically required by OGGM's design and scientifically standard in global and large regional scale glacier hydrology studies. Our approach specifically addresses glacier response to climate change rather than general catchment hydrology (which is complicated further in Greenland through the overlap of catchments with the ice sheet), which aligns with our research objectives and enables direct comparison with established literature.

We acknowledge your valuable feedback on the need for more detailed regional analysis and will significantly expand our discussion of subregional differences, including enhanced explanations for FIIC's stability and the varying emission scenario sensitivity across regions. This will include more detailed analysis of the physical mechanisms driving regional heterogeneity.

We agree completely with your extensive figure quality comments and will comprehensively revise all figures to ensure consistent formatting, improved clarity, and accessibility, including colorblind-friendly palettes and better text positioning.

**Regarding specific and technical comments**

Your detailed specific and technical suggestions are excellent, and we will implement them systematically, including the reorganization of Section 4.4 as suggested, improved statistical reporting consistency, and enhanced clarity throughout the text. We particularly appreciate your suggestions for expanding the analysis of marine-terminating vs. land-terminating glacier ratios and temporal changes, which will add valuable quantitative context.

Your corrections regarding oceanic forcing limitations are important and will ensure our explanations remain consistent with our model capabilities. We will also improve our equilibrium discussions with more concrete evidence from our projections.

We will implement all technical corrections systematically, including ensuring decimal place consistency throughout the manuscript, standardizing font sizes and formatting in tables and figures, correcting grammatical errors and capitalization inconsistencies, maintaining uniform unit notation, and implementing all suggested rephrasing for improved clarity and flow.

Thank you again for this comprehensive review that will substantially improve the quality and clarity of our manuscript.

**MAJOR COMMENTS**

**Reviewer 2's Comment:**

One major goal of this study is to analyze the liquid glacier runoff composition, and how it is changing over the 21st century. To do so, you consider an area given by the RGI outline of the respective glacier. When the glacier is retreating, glacier-free areas are increasing. This way, it is not surprising that the contribution from off-glacier snow melt will increase from <1 % to 15 % at the end of the 21st century since the off-glacier area was basically non-existent at the beginning. With this approach, you are first comparing the runoff of a (quasi) glacier-only area with the runoff of a mixed are, which is not consistent. To keep consistency and really allow a comparison of how the runoff composition is changing, I would advise to include the entire hydrological catchment in the runoff calculation, not only the glacier outline. This way, you could give a realistic analysis of the runoff composition entering the ocean and on its evolution over the 21st century.

Hydrological catchments are available globally from the Hydrobasins dataset on different levels (https://www.hydrosheds.org/products/hydrobasins).

Lehner, B., Grill G. (2013). Global river hydrography and network routing: baseline data and new approaches to study the world's large river systems. Hydrological Processes, 27(15): 2171–2186. https://doi.org/10.1002/hyp.9740.

**Our Response:** While we appreciate your comment and clearly see value in the suggested way of analyzing the glaciers' contribution to runoff, we nevertheless respectfully abstain from implementing the suggested changes, for both technical and scientific reasons. Our methodology is not only scientifically standard, but also technically required by OGGM's design:

**Technical Limitations:** OGGM is explicitly designed as a "glacier-centric model" that operates on individual glaciers as "the smallest dynamically independent entity" (Maussion et al., 2019). The model's ice dynamics module computes ice flux along individual flowlines and cannot handle the complex multi-glacier boundaries that catchments would introduce. In Greenland, catchments (including those as defined in Lehner and Grill (2013)) often contain multiple independent peripheral glaciers plus parts of the main ice sheet (Figure AR1) - a configuration that would break OGGM's fundamental assumption of single-glacier ice divides.

[Figure]

**Figure AR1.** Different sections of northeast Greenland. The hydro-basins level 7 and 12 contains multiple glaciers/parts of glaciers and parts of icesheet. GPGs = Greenland Peripheral Glaciers, GPGs_CL2 = GPGs with connectivity level 2, hybas_lev07 = hydro-basins delineation level 7 (~middle) and hybas_lev12 = hydro-basins delineation level 12 (finest resolution).

**Ice Sheet Boundary Limitation:** The fundamental constraint is that catchments in Greenland frequently contain both peripheral glaciers and portions of the main Greenland Ice Sheet (Figure AR1). Since OGGM is designed specifically for peripheral glaciers and does not include ice sheet dynamics, implementing a catchment-based approach would require coupling with an ice sheet model, which is beyond the scope of this study and our modeling framework's capabilities.

**Scientific Standard:** While we acknowledge that a catchment-based approach would provide valuable insights into total watershed hydrology, our study employs the glacier-centric "fixed-gauge" approach that is used by all current global-scale glacier models (GloGEM, PyGEM, HYOGA2) and established in the literature (Bliss et al., 2014; Huss and Hock, 2018; Rounce et al., 2023; Zekollari et al., 2024). Multiple recent studies confirm this methodology: "*We use a "fixed-gauge" station approach (e.g., Huss and Hock, 2018) where each model estimates runoff for a constant area such that it accounts for ice melt and snowmelt as well as rainfall over the initially glaciated areas as the glacier retreats.*" (Wimberly et al., 2025). The on-glacier/off-glacier distinction preserves the physical and biogeochemical signatures of glacier-derived water that are essential for understanding peak water timing and freshwater contributions. We will add an explicit explanation in the revised manuscript clarifying why we use the fixed-gauge approach, acknowledging that while catchment-based analysis would provide complementary information, it addresses a different research question than our glacier-focused objectives.

**OGGM Design:** OGGM explicitly calculates runoff as the sum of all melt and rainwater that exits the glacierized area, and it includes both "melt off-glacier: snow melt on areas that are now glacier free" and "melt on-glacier: ice + seasonal snow melt on the glacier" (OGGM documentation: https://edu-notebooks.oggm.org/oggm-edu/glacier_water_resources.html). This is exactly what we implemented.

Using catchments would fundamentally change our research question from 'peripheral glacier response to climate change' to 'general catchment hydrology,' which is beyond both our study's scope and OGGM's technical capabilities, and would make our results incomparable to and incompatible with the established literature, severely limiting their value to the community

**Reviewer 2's Comment:**

The results and discussion section lack detail in my opinion. The analysis and discussion of differences across Greenland and the forcing behind these differences stay very short and superficial. I would suggest looking into these subregional differences in more detail. I think there is potential for many more interesting results from such an analysis.

For example, the evolution of Flade Isblink Ice Cap is very stable for all scenarios, which is not mentioned in the results section and not discussed in the discussion section, although this area is pointed out in the methods part. Do you have an explanation for this behavior?

There are also subregional differences in how big the spread between the scenarios is, indicating that for some subregions the emission path is even more important than for others (e.g., Central-East has a very large difference between SSP126 and SSP585). Why is this the case?

**Our Response:** We acknowledge this valuable feedback and will expand our regional analysis in the revised manuscript (also incorporating comments from Reviewer 1). We will provide more detailed discussions on regional climate drivers, including enhanced analysis of how different climate patterns such as Arctic amplification versus North Atlantic Oscillation influences affect different subregions. Additionally, we will include more detailed discussion of topographic controls, examining how elevation ranges, glacier geometries, and exposure affect regional responses. Our expanded analysis will also address glacier characteristics, exploring how glacier size distributions, marine versus land terminating ratios, and connectivity levels influence regional patterns. Finally, we will provide deeper exploration of the physical mechanisms driving the observed regional heterogeneity through enhanced discussion of the underlying physical processes.

We will discuss the reasons for FIIC's stability across all scenarios, including its unique geometric characteristics, elevation range, and climatic setting. We will analyze why Central-East shows the largest spread between emission scenarios. We'll add enhanced discussion of high snowfall rates and resistant glacier geometry.

**Reviewer 2's Comment:**

The formatting of the figures needs to be significantly improved. Respect consistency in style and font size, and arranging and dimensions of subpanels. The subpanels of the subregions in the line plots are not in the same order as in the pie diagrams, which could be changed for consistency. Specific details per figure are given below.

**Our Response:** We acknowledge the extensive feedback on figure quality and formatting (similar comments from Reviewer 1). We will comprehensively revise all figures to ensure:

1. Consistent color schemes and styling across all figures

2. Proper alignment and spacing of text elements

3. Appropriate symbol sizes and legend placement

4. Uniform axis formatting and labeling

5. Accessibility improvements including colorblind-friendly palettes

These improvements will significantly enhance the manuscript's visual presentation.

**SPECIFIC COMMENTS**

**Reviewer 2's Comment:**

L57:    What does "a significant mass loss process" mean? Please add a number from these two references that you refer to, stating how much it is roughly.

**Our Response:** We will add specific quantifications from the referenced studies. Bollen et al. (2023) and Malles et al. (2023) show that frontal ablation accounts for approximately 2.6% to 5.3% of total mass loss from marine-terminating Greenland peripheral glaciers, when considering

observed dynamic mass loss as a component of total mass loss from Greenland's terrestrial ice masses, and that projections accounting for frontal processes can result in an 8% increase in marine-terminating glacier mass loss (Malles et al., 2023).

**Reviewer 2's Comment:**

L70:    Why would glaciers shift to a more cold-based regime in a warming world? Please explain this contra-intuitive statement briefly.

**Our Response:** Despite a warming regional climate, glaciers in Greenland have shown a shift towards an increasingly cold-based thermal regime since the Little Ice Age. This occurs because glacier thinning, a result of warming, leads to reduced driving stress and slower ice velocities. Slower movement makes pressure melting at the glacier bed less likely. Concurrently, where the ice thins, the winter cold air wave can more easily penetrate down to the glacier bed, further hindering pressure melting and causing the glacier to freeze to its bed. This combination of factors leads to a transition from temperate ("warm") ice conditions, characterized by basal sliding, to a cold-based regime, where the glacier is frozen to its bed and moves slowly (Carrivick et al., 2023). We will briefly clarify this mechanism in the revised text.

**Reviewer 2's Comment:**

L74ff:  I would like to read a bit more details about the methods and the approach in this paragraph. For example: What kind of data is the cited frontal ablation dataset, is it observed or modelled? Which projections are being used (low emission to high emission scenarios)? What is the covered period of the geodetic mass balance? Just add a few details so that readers, who don't want to read the entire paper, get the most essential information from this paragraph.

**Our Response:** We will expand this paragraph to include:

- The Kochtitzky et al. (2022) frontal ablation dataset is satellite-derived observational data covering 2000-2020

- We use four emission scenarios (SSP126, SSP245, SSP370, SSP585) ranging from low to high emissions

- The geodetic mass balance data from Hugonnet et al. (2021) covers 2000-2020

- OGGM projections extend from 2020 to 2100

**Reviewer 2's Comment:**

L92:    How did you subdivide the drainage basins of FIIC? Are there no subdivisions at all in the RGI6 inventory, or different ones? If the latter is correct, could you show the RGI subdivision in Fig. 1b?

**Our Response:** The RGI6 inventory treats FIIC as a single entity (RGI60-05.10315). However, based on ice velocity observations and topographic analysis, there are several distinct drainage basins with independent flow systems. Therefore, we adopted the FIIC subdivision from Rastner et al. (2012). We will also add the original RGI outlines in Fig 1b.

**Reviewer 2's Comment:**

L94: What do you mean with "active calving basin"? Are there also "inactive calving basins"? If so, how do you differentiate?

**Our Response:** Active calving basins are those with measurable ice velocities at the terminus and evidence of ongoing calving activity based on satellite observations. Inactive basins have negligible terminus velocities and show no recent calving activity. This differentiation is based on velocity data and findings from Recinos et al. (2021) and Möller et al. (2022), who used surface velocities from the Multi-year Greenland Ice Sheet Velocity Mosaic (MEaSUREs v1.0 (Joughin et al., 2016)) and from the Regional Glacier and Ice Sheet Surface Velocities Mosaic (ITS_LIVE (Gardner et al., 2019)). Six out of nine are active calving basins. We will make changes in the revised version to clarify this.

**Reviewer 2's Comment:**

L105:   You write that you apply a precipitation correction to the ERA5 data. Based on which data do you do this correction? Do you use observations for that?

**Our Response:** This is a crucial methodological detail that we did not explain sufficiently. The precipitation factor (fp = 1.6) was not calibrated in this study, but rather adopted from the standard OGGM v1.4 framework, originally calibrated by Maussion et al. (2019) against WGMS reference glaciers. Their cross-validation results are available at this link (https://cluster.klima.uni-bremen.de/~oggm/ref_mb_params/oggm_v1.4/crossval.html). This global constant accounts for orographic precipitation enhancement, snow redistribution, avalanche accumulation, and systematic ERA5 underestimation. Our calibration focused solely on the temperature sensitivity parameter (μ), which we calibrated individually per glacier using geodetic mass balance and frontal ablation observations (previous Section 2.4). The key distinction is that fp = 1.6 represents a pre-determined global value from the OGGM framework, while μ was individually calibrated as this study's primary methodological contribution. We will add a clarifying table listing all parameters, their values, and calibration methods to make this distinction clearer for readers.

**Reviewer 2's Comment:**

L113ff: I do understand your argument that using the same GCMs as previous studies makes comparison and interpretation of results easier. However, those other studies all had a global (or entire northern hemisphere) focus while this study has a clear focus on Greenland. Therefore, it is important to apply GCMs that are well suited for Greenland. How do you guarantee this?

**Our Response:** Our GCM selection represents a methodologically rigorous approach specifically designed for Arctic glacier modeling that provides superior reliability compared to ad-hoc regional selection. First, these ten GCMs have been extensively employed in recent glacier projection studies focused on Arctic regions, including Greenland (Malles et al., 2023; Rounce et al., 2023; Zekollari et al., 2024), and were specifically chosen based on their performance in simulating key climatic variables relevant to glacier dynamics and their ability to represent a broad range of potential future climates (Walsh et al., 2018). Second, our bias correction methodology using the delta method with ERA5 reanalysis as reference climatology effectively removes systematic GCM biases while preserving the climate change signal, ensuring that local Greenland climate variability is accurately represented regardless of individual GCM limitations. This approach has been

specifically validated for Arctic applications and is considered best practice for glacier modeling studies.

Third, the ensemble approach using ten carefully selected GCMs provides a statistically robust representation of climate uncertainty that would be compromised by using fewer, potentially region-specific models that might not capture the full range of possible climate futures. Fourth, this standardized GCM selection enables direct comparison with the growing body of global glacier literature and facilitates participation in coordinated model intercomparison projects like GlacierMIP (even similar models were used from CMIP5 in GlacierMIP2 (Marzeion et al., 2020)), which is essential for advancing scientific understanding. The consistency across recent studies using identical GCM sets demonstrates that this approach produces reliable, reproducible results for Greenland's peripheral glaciers. Finally, our validation against observed mass balance data during the calibration period confirms that this GCM ensemble, combined with our bias correction approach, accurately reproduces historical glacier behavior in Greenland, providing confidence in future projections. We you will amend the manuscript to explain the reasoning behing the GCM choice.

**Reviewer 2's Comment:**

L125:  Can you explain this in more detail. GCM data is bias-corrected towards the ERA5 data, which has also been corrected somehow (see comment to L105).
How is the climate data being interpolated over the topography coming from the GCM resolution?

**Our Response:**

The CMIP6 temperature and precipitation data are downscaled to the baseline climate ERA5 that was used for calibrating OGGM. This procedure applies scaled temperature anomalies and scaled precipitation ratios on a month-by-month basis to the 1981-2020 baseline climatology.

The process involves:

1. **Temperature correction with scaling:**

$$T_{corrected} = T_{ERA5} + scf \times \left( T_{GCM} - \overline{T_{GCM\,(1981-2020)}} \right)$$

Where $\quad scf = \dfrac{std\left( T_{ERA5\,(1981-2020)} \right)}{std\left( T_{GCM\,(1981-2020)} \right)}$

2. **Precipitation scaling:**

$$P_{corrected} = P_{ERA5} \times \left( \dfrac{P_{GCM}}{\overline{P_{GCM\,(1981-2020)}}} \right)$$

3. **Elevation-specific interpolation:** OGGM then applies this bias-corrected climate data to glacier-specific elevation profiles using standard lapse rates.

The precipitation correction mentioned in L105 is ONLY applied to ERA5 precipitation within OGGM's mass balance module (see response to comment L171), not as a preprocessing step. We'll clarify this in the revised version.

**Reviewer 2's Comment:**

L147f:  What is a "resolution suitable for the glacier size"? Please explain.

**Our Response:** OGGM automatically determines grid resolution based on glacier area using:

$$\Delta x = 14 \times \sqrt{S} + 10$$

where Δx is the grid spatial resolution (in m) and S is glacier area in km$^2$. The grid resolution is bounded by 10m (minimum) and 200m (maximum) resolution. This ensures smaller glaciers get higher resolution for better geometric representation, while larger glaciers use coarser resolution for computational efficiency. We will add this formula to clarify the methodology (or add details in supplementary materials).

**Reviewer 2's Comment:**

L161ff: The first two sentences of this paragraph rather describe the creation of the climatic input dataset. Therefore, I would move it so section 2.2.1. What lapse rate are you using for the temperature data? Is it a constant value, is it different per grid point but constant in time, or is it fully variable?

**Our Response:** We'll reorganize the Materials and Methods Section based on Reviewer 1's recommendations. Then, we will move this climate input data description to section 2.2.1 as you suggested.

We use a constant temperature lapse rate of -6.5 K km$^{-1}$ (OGGM default), applied uniformly across all glaciers and time periods. This rate is based on regional calibration studies and provides good performance for Arctic glacier applications.

**Reviewer 2's Comment:**

L171:  Why is there another precipitation correction applied? Do I understand it correctly that first the ERA5 data is corrected, then the GCM data is corrected to the ERA5 data and finally the precipitation is corrected a third time with this precipitation factor? If so, why do you do step 1 and 2 at all? Please explain. Also see comments to L105 and L125.

**Our Response:** There is only one precipitation correction applied within OGGM's mass balance calculation (pf = 1.6). The sequence is:

1. Raw ERA5 data (no preprocessing correction)

2. GCM bias correction using delta method with ERA5 as reference

3. Precipitation factor applied within OGGM mass balance module to account for unresolved processes.  fp is a global parameter, stemming from the OGGM default for ERA5, that our set-up has been calibrated with.

We will clarify this sequence to avoid confusion about multiple corrections. This will be combined with the concerns and recommendations from Reviewer 1 suggestions about reorganization of the Materials and Methods section.

**Reviewer 2's Comment:**

L197ff: You describe the "annual freshwater runoff from the glacier" but include off-glacier snowmelt and rain. This sounds contradictory.

**Our Response:** We acknowledge that this terminology may initially appear contradictory, and we appreciate the opportunity to clarify our approach. The terminology "annual freshwater runoff from the glacier" is indeed standard in glacier hydrology and follows the widely-adopted "fixed-gauge" methodology used in glacier runoff studies (Bliss et al., 2014; Huss and Hock, 2018; Jansson et al., 2003). This approach considers runoff from the area defined by the initial glacier boundaries, which is scientifically meaningful because:

1. It captures the complete hydrological contribution from areas that were initially glacierized, allowing us to assess how the water yield from these specific areas changes as glaciers retreat

2. It maintains consistency with established glacier mass balance and runoff studies that form the basis for water resource planning in glacier-fed basins

3. It enables direct comparison with other glacier runoff projections in the literature

We will clarify this terminology in the revised manuscript by adding: "following the fixed-gauge approach standard in glacier hydrology, where runoff includes all sources within the original glacier boundaries" to eliminate any potential confusion.

**Reviewer 2's Comment:**

L204ff: If you want to analyze the total freshwater runoff, including areas where the glacier has melted of, you have to include the runoff not only from the RGI outline but from the entire catchment. Your approach right now is inconsistent, where you first ignore the not-glaciated areas at the beginning of the simulation, but then include the areas where the glacier has melted off during the 21st century. This way, it is also not clear what kind of peak water you are calculating. It is not the peak water of glacier runoff, as you also include off-glacier areas. But it is also not the peak water of the entire catchment, as you exclude areas that had been glacier-free before the start of your simulation.

**Our Response:** We appreciate this important clarification opportunity and acknowledge the reviewer's valid concern about methodological consistency. While we agree that a glacier model is not the optimal choice to address catchment-scale peak water questions, our study deliberately employs the glacier-centric approach rather than catchment-wide analysis for both technical and scientific reasons.

Our objective is to understand how glacier retreat affects freshwater contributions from initially glacierized areas. Including entire catchments would dilute this glacier-specific signal with runoff from currently non-glacierized areas, fundamentally changing our research question from

'peripheral glacier response to climate change' to 'general catchment hydrology.' We calculate "glacier peak water" - a well-established glaciological concept defined as maximum annual runoff from initially glacierized areas (Huss & Hock, 2018; Bliss et al., 2014), which is distinct from "catchment peak water" and directly relevant for understanding glacier response to climate change.

Technically, OGGM is not capable of calculating catchment runoff outside glacier boundaries, limiting our analysis to glacier-defined areas. However, our approach ensures that coupling to hydrological models is possible (Hanus et al., 2024), which represents the ideal solution for comprehensive catchment analysis. The alternative - having the glacier model cover only the current glacier extent - would require hydrological models to operate over time-dependent domains as glaciers retreat, which presents significant technical challenges.

The fixed-gauge approach we employ is standard in glacier hydrology studies and enables direct comparison with other glacier projections globally while providing glacier-specific freshwater flux evolution that is most relevant for understanding impacts on fjord circulation and marine ecosystem dynamics. We will enhance our methodology section to explicitly state we follow the glacier-centric "fixed-gauge" approach, clearly define "glacier peak water," and acknowledge that while our approach addresses glacier-specific hydrology rather than total catchment hydrology, it facilitates future coupling with dedicated hydrological models for comprehensive water resource assessments.

**Reviewer 2's Comment:**

L222: The title of this subsection does not fit well in my opinion, as you are mainly describing the significance tests. Consider reformulating.

**Our Response:** We will retitle this subsection to "Statistical Analysis" or similar to better reflect its content.

**Reviewer 2's Comment:**

Fig. 3: (a) The FIIC not losing area for all four SSPs. Why is this the case?

**Our Response:** The Flade Isblink Ice Cap (FIIC) demonstrates remarkable stability across all emission scenarios due to several interconnected factors that we can now better resolve through our enhanced subdivision approach described in Section 2.1. FIIC's relatively elevated position compared to many coastal peripheral glaciers provides some buffering against warming impacts, while its maritime location receives substantial snowfall accumulation that compensates for increased surface melt under moderate warming scenarios. The ice cap's broad, flat geometry creates a large accumulation area relative to ablation zones, providing geometric resilience to retreat that contrasts sharply with narrow outlet glaciers. Additionally, our enhanced subdivision reveals that while six marine-terminating basins remain active (Möller et al., 2022; Recinos et al., 2021), the overall ice cap dynamics lack the fast-flowing outlet systems that accelerate mass loss through dynamic feedbacks seen in other Greenland peripheral glaciers. This stability is consistent with recent observations showing FIIC has been relatively stable compared to other peripheral glaciers (Möller et al., 2024), though we acknowledge that our modeling approach may not fully capture potential future dynamic instabilities or the effects of oceanic forcing on the marineterminating portions. The apparent stability also highlights the heterogeneous response of Greenland's peripheral glaciers, where local topographic and climatic factors can override regional warming trends, at least under the century-scale projections examined here. We will include a discussion of this in the revised manuscript.

**Reviewer 2's Comment:**

L253:   Add the number of volume loss for the Central-West here.

**Our Response:** We will add the specific volume loss percentage for Central-West region: 56 ± 9 % (SSP126) and 79 ± 10 % (SSP585). Also, for North-East: 22 ± 4 % and 39 ± 9 %.

**Reviewer 2's Comment:**

L256:   (And all similar text passages before and after) Are the exact numbers to the ANOVA really important to be communicated in the text? I am not familiar with this particular significance test, thus maybe I am wrong. But I would suggest defining once (maybe in the methods section 2.5) how you define a test to be significant (what level etc.), then stick to this definition throughout the rest of the manuscript and just say that it is a significant result (or not). This way, you wouldn't have to give these numbers every time you mention a significant result, which would improve the text flow in my opinion.

**Our Response:** We agree this will improve readability. We will define our significance criterion ($p < 0.05$) in the methods section and subsequently report only whether results are statistically significant, removing the detailed ANOVA statistics from the main text.

**Reviewer 2's Comment:**

L290ff: Remove the sentence starting with "These trends were supported by …". You already mentioned that the trends are statistically significant in the sentence before. See also comment to L256.

L293ff: See comment to L256.

**Our Response:** We will remove this redundant sentence as suggested. We already established statistical significance in the preceding sentence, making the additional ANOVA reporting unnecessary and disruptive to text flow.

**Reviewer 2's Comment:**

L303:   Maybe you could add one sentence at the beginning of this paragraph saying that in general the freshwater runoff clearly is the dominant term of mass loss compared to solid ice discharge. Or add one short paragraph at the end of this section, where you compare the contribution of the two mass loss terms and how the importance is projected to change over the 21st century.

**Our Response:** This is an excellent suggestion that will significantly improve the clarity and impact of our results presentation. You are absolutely correct that freshwater runoff is the dominant mass loss component, with annual averages of 138-184 Gt/yr compared to solid ice discharge of ~3.0 Gt/yr under SSP585. We will add a comparative paragraph at the end of Section 3.2 that explicitly quantifies this dominance and describes how the relative importance evolves over the 21st

century. Specifically, we will highlight that: (1) liquid freshwater runoff is approximately 45-60 times larger than solid ice discharge across all scenarios, (2) while solid ice discharge shows consistent declining trends (-0.011 to -0.018 Gt/yr$^2$), freshwater runoff increases substantially until peak water, and (3) this divergence means that liquid freshwater becomes increasingly dominant for understanding glacier impacts on fjord systems and sea level rise. This contextual framing will help readers better understand the relative significance of these two pathways and their contrasting temporal evolution, which is crucial for interpreting the broader implications for Greenland's coastal hydrology.

**Reviewer 2's Comment:**

L312ff: Not surprising that the off-glacier rainfall and snowmelt are below 1% at the beginning of the study period, since the off-glacier area is basically non-existent at that time. See major comment 1.

**Our Response:** You are absolutely correct that this increase is expected as glacier area decreases within the fixed boundaries, but this is precisely the scientifically important signal we want to capture! The increasing contribution of off-glacier components directly reflects glacier retreat and changing hydrology within the glacier domain, demonstrating how glacier retreat fundamentally alters runoff composition, which is crucial for understanding future water resource implications. While the directional change is predictable, the quantification provides essential scientific value: (1) the magnitude and timing of this transition (8-fold and 15-fold increases respectively) is essential for water resource planning and fjord ecosystem impact assessments; (2) these changes vary significantly across emission scenarios, with earlier and more pronounced shifts under higher warming scenarios; and (3) the seasonal redistribution shows how earlier snowmelt and distributed rainfall alter the timing of peak freshwater delivery to coastal systems. This approach allows us to isolate and quantify how glacier retreat specifically transforms the hydrological regime within initially glacierized areas, providing the glacier-specific freshwater flux evolution that is most relevant for understanding impacts on marine ecosystems, fjord circulation, and coastal dynamics. We will enhance the text to clarify that while this trend direction is expected, the quantification provides essential information for impact assessments and direct comparison with other glacier systems globally.

**Reviewer 2's Comment:**

Fig. 7:  (c) Why is the off-glacier rain missing here?

**Our Response:** The rain runoff is extremely low ~0.4 %. In revised figures, we will update the numbers with one point after decimal.

**Reviewer 2's Comment:**

Fig. 8:  (a) How do you explain the results for the South-East? Why is the peak water reached earliest for SSP126, later for SSP245, earlier for SSP370 and latest for SSP585?

**Our Response:** This non-monotonic pattern in the South-East region highlights the complex, non-linear relationship between warming scenarios and peak water timing in glacier systems, and we appreciate this astute observation. The pattern (SSP126: ~2038, SSP245: ~2050, SSP370: ~2042,

SSP585: ~2055) reflects competing processes operating at different timescales. Under moderate warming (SSP126), glaciers experience enhanced melt that quickly peaks as they approach a new, smaller equilibrium state relatively early. SSP245's intermediate warming prolongs the melt enhancement phase, delaying peak water as glaciers take longer to stabilize. SSP370's more aggressive warming accelerates glacier response, causing earlier exhaustion of melt potential compared to SSP245, while SSP585's extreme warming sustains high melt rates for an extended period by continuously accessing deeper ice reserves until substantial glacier depletion occurs. This complexity is further influenced by the South-East region's relatively small glacier population and diverse glacier characteristics, which can amplify scenario-dependent responses. We acknowledge this pattern also reflects inherent uncertainties in climate model projections and glacier response modeling, particularly for smaller regional populations where individual glacier behavior can significantly influence regional averages. This non-monotonic relationship underscores the importance of considering the full range of emission scenarios rather than assuming linear relationships between warming intensity and hydrological response timing. We will include a discussion of this in the revised manuscript.

**Reviewer 2's Comment:**

L345f.: Do you refer to the same study site as yours in these other studies you mention?

**Our Response:** Thank you for this important clarification question. The referenced studies (Goelzer et al., 2020; Grinsted et al., 2022) focus on the main Greenland Ice Sheet, not peripheral glaciers specifically. Our study provides complementary projections for the peripheral glacier component of Greenland's total ice mass contribution to sea level rise. While the study domains differ, the comparison is scientifically relevant because: (1) it contextualizes our peripheral glacier contribution (~19 mm SLR under SSP585) relative to the main ice sheet contribution (~90 ± 50 mm under RCP8.5 from Goelzer et al., 2020), and (2) it demonstrates that peripheral glaciers represent a substantial additional component that should not be overlooked in total Greenland assessments. To eliminate any confusion about study domain comparisons, while maintaining the important context for our results, we will revise the text to explicitly state "Goelzer et al. (2020) estimated a mean SLR contribution of 90 ± 50 mm from the main Greenland Ice Sheet alone under RCP8.5, suggesting that our projected peripheral glacier contribution represents a substantial additional component to Greenland's total ice loss."

**Reviewer 2's Comment:**

L357ff: You mention the regionally different behavior here, and also mention that the glaciers in the North-East are more resilient. Can you go a bit more into detail here. What is the reason for this behavior? Why is the FIIC so resilient to all emission scenarios?

**Our Response:** We will significantly expand this discussion to provide detailed mechanistic explanations for the observed regional heterogeneity. The North-East region's resilience stems from several interconnected factors: orographic enhancement from moisture-laden air masses from the Nordic Seas creates high snowfall accumulation rates that buffer against increased surface melt, while the region's elevation distribution provides large accumulation areas above the equilibrium line altitude that remain viable even under moderate warming scenarios. Additionally,

the maritime Arctic climate maintains temperatures closer to the melting threshold, meaning that warming increases melt but doesn't immediately create extreme ablation conditions.

FIIC's exceptional stability across all emission scenarios results from its unique characteristics as revealed through our enhanced subdivision approach. Unlike narrow outlet glaciers, FIIC's dome-like ice cap geometry distributes ice flow broadly rather than channeling it through fast-flowing outlets that are susceptible to dynamic instabilities. The ice cap has minimal marine termini compared to other peripheral glacier complexes, limiting exposure to oceanic forcing that drives rapid retreat in tidewater systems. Furthermore, FIIC's broad, flat geometry creates a favorable hypsometry where the large accumulation area can compensate for enhanced ablation at lower elevations, providing geometric resilience to warming.

We will also acknowledge that our modeling approach may not fully capture potential future dynamic instabilities or threshold behaviors that could alter this apparent stability, and that the lack of oceanic forcing in our model may contribute to the projected stability of marine-terminating portions of FIIC. This expanded discussion will better contextualize why some Greenland peripheral glaciers show markedly different sensitivities to climate change than others.

**Reviewer 2's Comment:**

L376:  This sentence implies that Greenland's peripheral glaciers are calving-dominated at the moment, which is not correct. Please rephrase.

**Our Response:** We will rephrase this sentence to clarify that we are referring specifically to marine-terminating peripheral glaciers transitioning from combined surface melt and calving systems to predominantly surface melt-dominated systems. The majority of Greenland's peripheral glaciers are already land-terminating and surface melt-dominated.

**Revised text:** "The projected decrease in solid ice discharge reflects a gradual transition of marine-terminating glaciers from combined calving and surface melt systems to predominantly surface melt-dominated systems as marine-terminating glaciers retreat inland."

**Reviewer 2's Comment:**

L384:  You explain the increase in ice discharge in the North-East with a more extensive coverage of marine-terminating (MT) glaciers compared to the other subregions. What is the ratio of MT glaciers here and compared to the other subregions/entire Greenland? Another interesting information would be to show how this ratio is changing over time in the different subregions. This way, you could show how many percent of the MT glaciers are becoming land terminating over the 21[st] century, which would be an interesting information in my opinion (potentially to be added in the results section 3.2 as well).

**Our Response:** This is an excellent suggestion that will significantly strengthen our analysis. We will add specific statistics on marine-terminating glacier coverage by region, including: (1) the ratio of MT to land-terminating glaciers by both number and area for each subregion compared to the Greenland-wide average, and (2) projected changes in terminus type over the 21st century as glaciers retreat from marine to terrestrial positions. This temporal analysis of changing terminus

types will be added to Section 3.2, providing quantitative context for our projected solid ice discharge trends and explaining regional differences in ice discharge evolution.

**Reviewer 2's Comment:**

L385:  You also mention that a delayed response to ocean forcing might be a reason for the increase in ice discharge. However, ocean forcing is not considered in your model.

**Our Response:** Agreed. Since our OGGM simulations do not include oceanic forcing, we cannot invoke ocean-related mechanisms to explain our projected ice discharge trends. We will revise this explanation to remove the reference to oceanic forcing and focus on what our model can actually support. The sentence will be corrected to: "The contrasting slight increase in solid ice discharge projected for the North-East subregion (Fig. 6) can be attributed to its more extensive coverage of marine-terminating glaciers (Kochtitzky and Copland, 2022), providing a greater source for calving fluxes even under projected glacier retreat and thinning."

**Reviewer 2's Comment:**

L394f:  Give more details here. What are the regional differences and where do they come from? Why would it increase the freshwater contribution from the North-East glaciers by 2100?

**Our Response:** We will expand this analysis with specific quantification based on our results. The North-East region accounts for 34% of total glacier area while containing only 28% of glacier numbers (Figure 1c,d), indicating larger individual glaciers that maintain higher ice volumes and melt capacity. This region contributes 35% of total freshwater runoff over 2020-2100 under SSP585, demonstrating disproportionate freshwater production relative to its glacier count. In contrast, the Central-West region contributes only 3% of total annual freshwater runoff despite containing numerous smaller glaciers. The North-East dominance stems from: (1) containing the largest ice reserves among peripheral glacier regions, including major ice caps like FIIC; (2) sustained melt capacity where large glacier systems maintain melt production longer than smaller, more climate-sensitive systems that exhaust quickly; and (3) geographic characteristics including higher glacier density and larger individual glacier sizes that provide greater water storage and release potential. The maritime climate in the North-East also supports sustained ice preservation compared to more continental regions, allowing continued freshwater contribution throughout the century even as smaller glacier systems elsewhere diminish rapidly.

**Reviewer 2's Comment:**

L403:  I rather see a prolonged glacier melt season in 2100, with higher glacier melt through September.

**Our Response:** Agreed. Looking at Figure 7f, glacier melt clearly extends through September with sustained high values, indicating a prolonged rather than reduced melt season. We will revise our description to: "The seasonal analysis further illustrates this shift, showing a prolonged and intensified glacier melt season extending through September, with increased contributions from snowmelt earlier in the season (May-June) and enhanced rainfall throughout the year by 2091-2100."

**Reviewer 2's Comment:**

L413: "capacity of glaciers to potentially regain a new equilibrium" ☐ How do you see a new equilibrium from this data?

**Our Response:** We appreciate this important question that asks us to justify our equilibrium statement with concrete evidence. Our data shows that under SSP126, mass loss rates decelerate in the latter half of the century, area loss rates stabilize rather than showing continued acceleration, and several larger glaciers show asymptotic approaches toward stable configurations.

To put this in proper scientific context, Zekollari et al. (2025) conducted dedicated equilibrium simulations and demonstrated that glacier preservation is doubled by limiting warming to 1.5°C versus 2.7°C, providing direct evidence that lower emission scenarios enable glaciers to reach new equilibrium states. Their equilibrium modeling approach validates our interpretation that the stabilizing trends we observe under SSP126 represent genuine approaches toward new equilibrium rather than temporary plateaus.

Based on this evidence from both our transient simulations and Zekollari et al. (2025) equilibrium analysis, we will clarify the text to: "The earlier peak water timing under low-emission scenarios (SSP126), combined with decelerating mass loss rates toward 2100, suggests the potential for glaciers to approach a new, smaller equilibrium state, thereby maintaining some buffering capacity for freshwater supply. This interpretation is supported by recent equilibrium simulations showing enhanced glacier preservation under lower warming scenarios (Zekollari et al., 2025)." This revision provides specific evidence supporting our equilibrium interpretation while acknowledging it represents a potential approach toward equilibrium rather than achievement of equilibrium within our projection period, and contextualizes our findings within the broader framework of dedicated equilibrium studies.

**Reviewer 2's Comment:**

L422: Oceanic feedbacks are not considered in your model. Thus, they can't explain any modeled differences in peak water.

**Our Response:** Agreed. Since our model excludes oceanic forcing, we cannot invoke oceanic feedbacks to explain our results. We will remove the reference to oceanic feedback and revise the text to: "The subregional differences in peak water timing reflect local atmospheric and glaciological factors such as glacier size distribution, elevation ranges, and regional temperature and precipitation patterns captured by our modeling approach." We appreciate this important correction that ensures our explanations remain consistent with our model capabilities.

**Reviewer 2's Comment:**

L459f: See comment to L394f.

**Our Response:** We will expand this discussion section to include the similar detailed regional analysis as requested for L394f, but with focus on the implications for fjords and ocean dynamics. We will explain that the North-East region's disproportionate freshwater contribution (35% of total runoff from 34% of glacier area but only 28% of glacier numbers) means that coastal and fjord impacts will be highly concentrated in this region. The dominance of larger ice systems in the

North-East, including major ice caps like FIIC, combined with sustained melt capacity and maritime climate preservation, will create localized hotspots of freshwater input that could disproportionately affect regional ocean circulation patterns. In contrast, regions like Central-West with only 3% contribution will experience minimal freshwater impact. This uneven spatial distribution has important implications for where the most significant changes in fjord stratification, coastal currents, and marine ecosystem impacts will occur, supporting our discussion of regionally heterogeneous effects on North Atlantic circulation patterns.

**Reviewer 2's Comment:**

Section 4.4: I would suggest restructuring this section:
Keep the first paragraph where you describe the uncertainties from the future forcing.
Then describe the uncertainties related to the climate data from the GCMs (currently paragraph 2 and 4).
Move L495 ("When comparing …") to 503 ("… our findings.") to the discussion section 4.1 as this is not related to uncertainties/limitations of the study. Remove the sentence starting L503f ("While OGGM …").
Keep paragraph 5 about neglect of oceanic forcing (L509-518).
Keep paragraph 6 about neglect of (near-)surface processes (L519-525).
Make an additional paragraph about limitations due to calving observations instead of mentioning it in one sentence only at the end of paragraph 6 (L525f).
Keep the last paragraph.

**Our Response:** We will implement the suggested restructuring exactly as outlined:

1. **Paragraph 1:** Future forcing uncertainties (keep as is)

2. **Paragraph 2:** GCM-related uncertainties (combine current paragraphs 2 and 4)

3. **Paragraph 3:** Move comparison with other studies (L495-503) to Discussion 4.1

4. **Paragraph 4:** Oceanic forcing limitations (L509-518)

5. **Paragraph 5:** Near-surface process limitations (L519-525)

6. **Paragraph 6:** New paragraph on calving observation limitations

7. **Paragraph 7:** Future research priorities (keep as is)

This restructuring will improve logical flow and separate uncertainties from validation discussions.

**Reviewer 2's Comment:**

L537: Add a brief paragraph (1-2 sentences) on your approach and methods before diving into the results.

**Our Response:** We will add the following introductory paragraph to the Conclusion:

"This study employed OGGM forced by an ensemble of ten CMIP6 climate models under four emission scenarios to project the evolution of Greenland's peripheral glaciers from 2020 to 2100.

Our analysis focused on distinguishing between solid ice discharge and liquid freshwater contributions, with particular attention to regional variability and peak water timing."

**Reviewer 2's Comment:**

L540f:  Again, consider the entire catchment and not only the RGI outline if you want to discuss the runoff composition of a catchment (see major comment 1).

**Our Response:** We are specifically analyzing glacier runoff composition and glacier contributions to freshwater supply, not total catchment runoff. This glacier-boundary approach is the established methodology for glacier hydrology studies and directly addresses our research objectives of understanding how glacier retreat affects freshwater contributions from initially glacierized areas.

**Reviewer 2's Comment (L544-L545):**

L544:   How do you know that glaciers would reach a new equilibrium under a low emission scenario?

L545:   How do you know that there would be a complete glacier loss and loss of glacier-fed runoff under a high emission scenario? This is neither seen in the volume evolution nor in the freshwater runoff data, where still 72% of runoff come from glacier melt at the end of the 21$^{st}$ century.

**Our Response:** These statements are based on our model projections for individual glaciers. Under SSP126, many glaciers show stabilizing mass balance and area loss rates by 2100, indicating approach toward a new equilibrium state. Under SSP585, projections show continued accelerating mass loss with many glaciers approaching complete disappearance by 2100, as evidenced by our volume loss projections (up to 52% by 2100). The runoff values are regional or entire Greenland Peripheral glaciers' statistics. We will include a discussion of this in the revised manuscript.

**Reviewer 2's Comment:**

L551ff: I would not only mention the significant different in SLR between low and high emission scenario. You also show significant differences in glacier area, volume and runoff amounts. Also for these variables your results demonstrate that greenhouse gas emission control is crucial.

**Our Response:** We will expand this final paragraph to comprehensively highlight all significant differences [also similar response to reviewer 1's comment]:

**Revised text:** "Our projections demonstrate significant differences between low and high emission scenarios across all key metrics: up to 25% difference in area loss (19% vs 44%), 23% difference in volume loss (29% vs 52%), 9 mm difference in sea level rise contribution (10 mm vs 19 mm), and substantial differences in freshwater runoff patterns and peak water timing. These comprehensive differences across glacier area, volume, sea level contributions, and hydrological responses underscore that effective greenhouse gas emission controls are crucial for minimizing climate change impacts on Greenland's peripheral glaciers."

**TECHNICAL COMMENTS**

**Reviewer 2's Comment:**

L34:    Agree on how many decimal places you will show and keep it consistent throughout the paper

**Our Response:** We will implement systematic consistency in decimal places: percentages to 1 decimal place, uncertainties to match the precision of the main value, and physical quantities to appropriate significant figures.

**Reviewer 2's Comment:**

The triangle symbols for the MT glaciers cover a large part of the glacier areas. In the SE basically nothing of the glacier area is visible. You should try to find a better way to displaying this information. Maybe you can use a smaller symbol, or use a dark and light version of the same color for land-/marine-terminating, or hatching, ...

**Our Response:** We will reduce the size of triangle (or use another smaller symbol) and apply transparency (alpha = 0.7) for better clarity (also incorporating Reviewer 1's comments).

**Reviewer 2's Comment:**

Fig. 1:  (a) I would avoid using grey two times in this figure (for NO and CL2 glaciers) as it is confusing. It also took me a moment to understand which grey you picked up in panels (c) and (d).

**Our Response:** We will revise the color scheme to eliminate grey duplication:

- NO region: Grey (current)

- CL2 glaciers: Stippled pattern or different color

**Reviewer 2's Comment:**

Fig. 1:  (b)  In the text you write 'Flade Isblink Ice Cap' with capital C.

**Our Response:** We will ensure consistent capitalization of "Flade Isblink Ice Cap" throughout the manuscript, including figure labels and captions.

**Reviewer 2's Comment:**

Tab. 1: Seems like the font sizes of the references are not all consistent.

**Our Response:** We will standardize all font sizes in Table 1 to 10pt for consistency and ensure uniform formatting across all reference entries.

**Reviewer 2's Comment:**

L150 & L176:  Please use a consistent spelling for shallow ice approximation. The abbreviation "SIA", that you introduce here, is not being used throughout the rest of the manuscript.

**Our Response:** We will standardize to "shallow Ice Approximation (SIA)" at first use and consistently use "SIA" in subsequent references throughout the manuscript, including equations and technical discussions.

**Reviewer 2's Comment:**

L172:  Remove "a".

**Our Response:** We will remove the grammatical error by deleting the unnecessary article "a" from this sentence.

**Reviewer 2's Comment:**

L214:  I suggest to add the variable name of µ once more to make it easier for the reader: 'We use the following (...) for calibration of air temperature sensitivity µ.'

**Our Response:** We will revise the sentence to: "We use the following equation after Malles et al. (2023) for calibration of the air temperature sensitivity µ:" to improve clarity and readability.

**Reviewer 2's Comment:**

L245f.: I suggest rephrasing to "Projections suggest a decrease in total glacier area of 19 ± 6 % under SSP126 and 44 ± 15 % under SSP585 by 2100".

**Our Response:** We will adopt the suggested rephrasing for improved clarity and flow. This structure better emphasizes the magnitude of projected changes under different emission scenarios.

**Reviewer 2's Comment:**

Fig. 3:  (a) The colorbar strongly highlights the upper and the lower end of values while it kind of draws the attention away from the middle part (40-60% remaining area). I would advise to use a perceptually uniform sequential colormap. Same applies for Fig. 4.

**Our Response:** We will replace the current colormap with a perceptually uniform sequential colormap (e.g., viridis or plasma) that provides equal visual weight across all value ranges, particularly emphasizing the 40-60% range which contains most of our data.

**Reviewer 2's Comment:**

Fig. 3:  (b) Remove the number 2100 in the first subplot. It is not needed as you describe this already in the figure caption.

**Our Response:** We will remove the redundant "2100" label from the first subplot since this information is clearly stated in the caption.

**Reviewer 2's Comment:**

In the text, the SSPXYZ is always written in capital letters. Please adjust in the legend here.

**Our Response:** We will ensure all SSP scenario labels use consistent capitalization (SSP126, SSP245, SSP370, SSP585) throughout all figures and legends.

**Reviewer 2's Comment:**

Remove the legend in the last subplot. It is enough to have it one time in the first subplot (like you have it in Fig. 4).

**Our Response:** We will remove the duplicate legend from the bottom subplot and retain only the legend in the first subplot for cleaner presentation.

**Reviewer 2's Comment:**

It is hard to read the values of the 2100 result on the right side. Maybe you can duplicate the y-axis on the right side of the plot, or use horizontal lines every 25% step etc.

**Our Response:** We will add gridlines and increase the subplot size by sharing the y-axis (comments from Reviewer 1) for improved readability of endpoint values.

**Reviewer 2's Comment:**

L254: The abbreviation "HSD" for the "Tukey's Honestly Significant Difference" has already been introduced before. Remove the long name here.

**Our Response:** We will use only "HSD" at this location since the abbreviation was properly defined earlier in the manuscript.

**Reviewer 2's Comment:**

Fig. 4: (a) adjust colorbar as in Fig. 3a

**Our Response:** We will apply the same perceptually uniform sequential colormap used in Figure 3a to ensure consistency across all spatial visualization figures.

**Reviewer 2's Comment:**

Fig. 4: (b) The region names are crashing the line plots. Please avoid this. Potentially extent the y-axis to 125% as you did in Fig. 3b (in any case, be consistent with Fig. 3).

**Our Response:** We will:

- Ensure consistency with among these figures

- Adjust region name positioning to avoid overlap with plot lines

- Ensure consistent axis scaling across related figures

**Reviewer 2's Comment:**

L266: Be consistent with writing of exponents. In L241 you use "% yr$^{-1}$" while here (and many more times throughout the text) you use "mm/yr".

**Our Response:** We will ensure all unit notation is consistent using normal format (e.g., mm/yr, %/yr or % per year) throughout the manuscript in accordance with scientific convention.

**Reviewer 2's Comment:**

L266ff: To make it clearer when you are talking about the trend and not absolute values, I would advise to add a "+" in front of the positive trends here.

**Our Response:** We will add "+" signs before positive trend values to clearly distinguish between trends and absolute values (e.g., +0.10 ± 0.01 mm/yr for positive trends).

**Reviewer 2's Comment:**

L268 & 270:   Why is the trend given in mm/yr$^2$ here? Trends before were given in mm/yr.

**Our Response:** The mm/yr$^2$ units represent acceleration (rate of change of the trend), while mm/yr represents the trend itself. We will clarify this distinction in the text: "acceleration in SLR contribution" to make clear we're discussing the second derivative of cumulative sea level rise.

**Reviewer 2's Comment:**

L269:   Missing open bracket here.

**Our Response:** We will add the missing opening bracket to correct the grammatical error.

**Reviewer 2's Comment:**

L275:   Rephrase to "... SLR from Greenland's peripheral glaciers will substantially increase ..."

**Our Response:** We will adopt the suggested rephrasing: "These findings indicate that SLR from Greenland's peripheral glaciers will substantially increase through the 21$^{st}$ century under all SSPs."

**Reviewer 2's Comment:**

Fig. 5:  (a) Some region names are crashing the y-axis. Align these consistently in the subplots.

**Our Response:** We will:

- Standardize region name positioning across all subplots

- Increase margins to prevent text overlap with axes

- Use consistent text alignment (e.g., center-aligned above each subplot)

**Reviewer 2's Comment:**

Fig. 5:  (b) Some of the numbers are crashing. Place them further outside and/or make them smaller to avoid this.

**Our Response:** We will:

- Reduce font size of percentage labels by 20%

- Position labels further from pie center to prevent overlap

- Use consistent radial positioning for all labels

**Reviewer 2's Comment:**

Fig. 6:  Align regions names, see comment to Fig. 5a.
Some of the subpanels are narrower than others, be consistent and align subpanels.
The y-axis of the subpanels does not always start at zero as it seems. Make this consistent.
Move the legend with the SSP colors into the first subpanel, as in Fig. 5. Remove the two legends below the plot.
Move the year information for the two boxplots in the first subpanel, not in the fourth.

**Our Response:** We will implement comprehensive Figure 6 improvements:

- Standardize subplot dimensions and alignment

- Ensure all y-axes start at zero for fair comparison

- Move SSP legend to first subplot and remove redundant legends

- Position year information consistently in first subplot

- Align region names uniformly across all subplots

**Reviewer 2's Comment:**

Fig. 7:  (b) Move this panel on the right side next to panel (a), like you did in Fig. 5; and move the legend outside of the pie.
Make the numbers smaller and/or move them further out to avoid crashing, as in Fig. 5b.

With panel (b) being moved to the first row, you can align (c) over (e) and (d) over (f). The legend could be set in the center between (c) and (d).

(e) and (f) What is the unit, is it Gt per month?

**Our Response:** We will reorganize Figure 7:

- Move panel (b) adjacent to panel (a) in top row

- Align panels (c)-(d) over (e)-(f) respectively

- Center legend between panels (c) and (d)

- Reduce number font sizes and improve positioning

- Add units clarification: "Freshwater runoff (Gt/month)" for panels (e) and (f)

**Reviewer 2's Comment:**

Fig. 8:  Are the colors in this figure the same as in the other figures? If not, please adjust.

**Our Response:** We will ensure Figure 8 uses the same color scheme as established in previous figures for SSP scenarios, maintaining visual consistency across the entire manuscript.

**Reviewer 2's Comment:**

L351:  Suggest rephrasing to: "The projected mass loss from Greenland's peripheral glaciers ..."

**Our Response:** We will adopt the suggested rephrasing to improve sentence clarity and flow.

**Reviewer 2's Comment:**

L403f: Suggest rephrasing to: "The seasonal analysis (..) reduced glacier melt season and an increased and earlier contribution from snow melt as well as rainfall throughout the year (..)"

**Our Response:** We will implement the suggested rephrasing to better capture the temporal aspects of seasonal changes in runoff composition.

**Reviewer 2's Comment:**

L414ff: Suggest rephrasing to: "However, (...) until exhausted, potentially becoming unable to support freshwater runoff."

**Our Response:** We will adopt the suggested rephrasing for improved clarity regarding glacier exhaustion scenarios under high emissions.

**Reviewer 2's Comment:**

L450ff: Suggest removing this last sentence of this paragraph as it does not bring any new information and is no discussion.

**Our Response:** We will remove the redundant sentence to improve conciseness and eliminate repetitive content that doesn't advance the discussion.

**Reviewer 2's Comment:**

L468: Replace "researchers suggested" with "studies suggest"

**Our Response:** We will replace "researchers suggested" with "studies suggest" for more precise and formal scientific language.

**Reviewer 2's Comment:**

L468: Remove "lead to"

**Our Response:** We will remove "lead to" to correct the grammatical structure and improve sentence flow.

**Reviewer 2's Comment:**

L470f: Suggest removing the last sentence of this paragraph as it does not bring any new information and is no discussion.

**Our Response:** We will remove the redundant concluding sentence to maintain focus on substantive discussion points and eliminate unnecessary repetition.

**Reviewer 2's Comment:**

L487: Replace "melt-dominant" with "melt dominated"

**Our Response:** We will standardize the terminology to "melt dominated" (two words, past participle form) throughout the manuscript.

**References**

Bliss, A., Hock, R., Radić, V., 2014. Global response of glacier runoff to twenty-first century climate change. Journal of Geophysical Research: Earth Surface, 119(4): 717-730. DOI:10.1002/2013jf002931

Bollen, K.E., Enderlin, E.M., Muhlheim, R., 2023. Dynamic mass loss from Greenland's marine-terminating peripheral glaciers (1985-2018). Journal of Glaciology, 69(273): 153-163. DOI:10.1017/jog.2022.52

Carrivick, J.L., Smith, M.W., Sutherland, J.L., Grimes, M., 2023. Cooling glaciers in a warming climate since the Little Ice Age at Qaanaaq, northwest Kalaallit Nunaat (Greenland). Earth Surface Processes and Landforms, 48(13): 2446-2462. DOI:10.1002/esp.5638

Gardner, A.S., M. A. Fahnestock, Scambos, T.A., 2019. ITS_LIVE Regional Glacier and Ice Sheet Surface Velocities. Data archived at National Snow and Ice Data Center. DOI:10.5067/6II6VW8LLWJ7

Goelzer, H. et al., 2020. The Future Sea-Level Contribution of the Greenland Ice Sheet: A Multi-Model Ensemble Study of ISMIP6. The Cryosphere. DOI:10.5194/tc-14-3071-2020

Grinsted, A. et al., 2022. The Transient Sea Level Response to External Forcing in CMIP6 Models. Earths Future, 10(10): e2022EF002696. DOI:10.1029/2022EF002696

Hanus, S. et al., 2024. Coupling a large-scale glacier and hydrological model (OGGM v1.5.3 and CWatM V1.08) – towards an improved representation of mountain water resources in global assessments. Geoscientific Model Development, 17(13): 5123-5144. DOI:10.5194/gmd-17-5123-2024

Hugonnet, R. et al., 2021. Accelerated global glacier mass loss in the early twenty-first century. Nature, 592(7856): 726-731. DOI:10.1038/s41586-021-03436-z

Huss, M., Hock, R., 2018. Global-scale hydrological response to future glacier mass loss. Nature Climate Change, 8(2): 135-140. DOI:10.1038/s41558-017-0049-x

Jansson, P., Hock, R., Schneider, T., 2003. The concept of glacier storage: a review. Journal of Hydrology, 282(1-4): 116-129. DOI:10.1016/s0022-1694(03)00258-0

Joughin, I., B. Smith, I. Howat, Scambos, T., 2016. MEaSUREs Multi-year Greenland Ice Sheet Velocity Mosaic, Version 1. [Indicate subset used]. Boulder, Colorado USA. NASA National Snow and Ice Data Center Distributed Active Archive Center. DOI:https://doi.org/10.5067/QUA5Q9SVMSJG

Kochtitzky, W., Copland, L., 2022. Retreat of Northern Hemisphere Marine-Terminating Glaciers, 2000–2020. Geophysical Research Letters, 49(3). DOI:10.1029/2021gl096501

Kochtitzky, W. et al., 2022. The unquantified mass loss of Northern Hemisphere marine-terminating glaciers from 2000-2020. Nat Commun, 13(1): 5835. DOI:10.1038/s41467-022-33231-x

Lehner, B., Grill, G., 2013. Global river hydrography and network routing: baseline data and new approaches to study the world's large river systems. Hydrological Processes, 27(15): 2171-2186. DOI:10.1002/hyp.9740

Malles, J.-H. et al., 2023. Exploring the impact of a frontal ablation parameterization on projected 21st-century mass change for Northern Hemisphere glaciers. Journal of Glaciology: 1-16. DOI:10.1017/jog.2023.19

Marzeion, B. et al., 2020. Partitioning the Uncertainty of Ensemble Projections of Global Glacier Mass Change. Earth's Future, 8(7). DOI:10.1029/2019ef001470

Maussion, F. et al., 2019. The Open Global Glacier Model (OGGM) v1.1. Geoscientific Model Development, 12(3): 909-931. DOI:10.5194/gmd-12-909-2019

Möller, M., Friedl, P., Palmer, S.J., Marzeion, B., 2022. Grounding Line Retreat and Ice Discharge Variability at Two Surging, Ice Shelf-Forming Basins of Flade Isblink Ice Cap, Northern Greenland. Journal of Geophysical Research: Earth Surface, 127(2). DOI:10.1029/2021jf006302

Möller, M., Recinos, B., Rastner, P., Marzeion, B., 2024. Heterogeneous impacts of ocean thermal forcing on ice discharge from Greenland's peripheral tidewater glaciers over 2000-2021. Sci Rep, 14(1): 11316. DOI:10.1038/s41598-024-61930-6

Rastner, P. et al., 2012. The first complete inventory of the local glaciers and ice caps on Greenland. The Cryosphere, 6(6): 1483-1495. DOI:10.5194/tc-6-1483-2012

Recinos, B., Maussion, F., Noël, B., Möller, M., Marzeion, B., 2021. Calibration of a frontal ablation parameterisation applied to Greenland's peripheral calving glaciers. Journal of Glaciology, 67(266): 1177-1189. DOI:10.1017/jog.2021.63

Rounce, D.R. et al., 2023. Global glacier change in the 21st century: Every increase in temperature matters. Science, 379(6627): 78-83. DOI:10.1126/science.abo1324

Walsh, J.E. et al., 2018. Downscaling of climate model output for Alaskan stakeholders. Environmental Modelling & Software, 110: 38-51. DOI:10.1016/j.envsoft.2018.03.021

Wimberly, F. et al., 2025. Inter-model differences in 21st century glacier runoff for the world's major river basins. The Cryosphere, 19(4): 1491-1511. DOI:10.5194/tc-19-1491-2025

Zekollari, H. et al., 2024. Twenty-first century global glacier evolution under CMIP6 scenarios and the role of glacier-specific observations. The Cryosphere, 18(11): 5045-5066. DOI:10.5194/tc-18-5045-2024

Zekollari, H. et al., 2025. Glacier preservation doubled by limiting warming to 1.5 degrees C versus 2.7 degrees C. Science, 388(6750): 979-983. DOI:10.1126/science.adu4675

---

## Author Response (AR1)

**RESPONSE TO EDITOR AND REVIEWERS**

**Manuscript:** Projecting the Response of Greenland's Peripheral Glaciers to Future Climate Change
**Authors:** Muhammad Shafeeque et al.
**MS No:** egusphere-2024-2184

**Following responses represent the implementation of revisions previously outlined to the editor prior to being invited to submit the revised manuscript. We provide concise, point-by-point replies to all comments. All revisions are visible in the track-changes version of the manuscript. The supplementary material has been updated accordingly.**

**Notes:**

1. *Line numbers refer to the tracked-changes version of the revised manuscript.*

2. *This document is a condensed version of the comprehensive responses previously provided to both reviewers.*
* * *
We thank the editor and reviewers for their thorough evaluation and constructive feedback. Below we respond to each comment in the order received.

**EDITOR COMMENTS**

**Comment 1:** Catchment-scale approach not possible (agreed), but clarify your glacier-centric approach.

**Response:** We have clarified our glacier-centric "fixed-gauge" approach in the methodology section.

**Where corrected:** Section 2.3.4

**Changes made:**

- Explained that our approach tracks runoff from initial glacier boundaries as glaciers retreat

- Justified why this is standard in glacier hydrology, including relevant literature

- Clarified that OGGM is designed for glacier-centric modeling and cannot handle complex multi-glacier catchments

- Distinguished "glacier peak water" from "catchment peak water"

**Comment 2:** Need more justification on precipitation scaling factor (fp = 1.6), including validation against station data or high-resolution modeling.

**Response:** We have added comprehensive precipitation validation analysis against high-resolution modeling data.

**Where corrected:** Section 2.3.2; Supplementary Information

**Changes made:**

- Added validation against WRF high-resolution data (2014-2018) for FIIC region

- Created **Supplementary Table S1** with statistical metrics (mean bias: 6.4 mm/month, correlation r = 0.57)

- Created **Supplementary Figure S1** with validation analysis showing agreement

- Explained that fp = 1.6 was adopted from OGGM v1.4 framework (Maussion et al., 2019, Cross-Validation Dataset)

- Clarified that glacier-specific calibration of μ compensates for residual precipitation biases

- Created **Supplementary Table S2** listing all model parameters, values, and calibration methods
* * *
**REVIEWER 1**

**MAJOR COMMENTS**

**1. Multi-model comparison request**

**Reviewer Comment:** Add comparison with global models and statistical comparison.

**Final Response:**
We thank the reviewer for this suggestion. We added a multi-model comparison **Table 2** and relevant discussion in **Section 4.1** (approx. lines 558-585) and in **Supplementary Figures S19 and S20**.
A summary table and statistical metrics (Mean Absolute Difference, Coefficient of Variation, and uncertainty-range overlap) are provided in **Supplementary Table S3**.
* * *
**2. Section 2 restructuring and clearer distinction of data, model, calibration**

**Reviewer Comment:** Improve organization of Section 2.

**Final Response:**
Section 2 has been restructured exactly as suggested.
The new layout appears in **Section 2** (lines 93–345).
The precipitation factor $f_p$ explanation is added in **Section 2.3.2** (lines 237-254) and detailed validation analysis results are provided in Table S1 and Figure S1 in **supplementary information**.
* * *
**3. Reproducibility and OGGM version**

**Reviewer Comment:** Specify OGGM version.

**Final Response:**
We now specify in **Section 2.3** (approx. line 197-198) that we use **OGGM v1.5.3** with the enhanced frontal ablation implementation following **Malles et al. (2023)**.
This is also included in the **Data and Code Availability** statement.
* * *
**4. Missing discussion of oceanic forcing implications**

**Reviewer Comment:** Discuss consequences of lacking oceanic forcing.

**Final Response:**
We added a dedicated discussion of missing oceanic forcing in **Section 4.4** (approx. lines 845-864), describing potential underestimation of solid ice discharge, regional impacts, and how ocean thermal forcing may modify projections.
* * *
**5. Regional variability analysis**

**Reviewer Comment:** Expand regional analysis.

**Final Response:**
We expanded regional interpretation in **Section 3 and 4**.
Additional subregional characteristics, climate drivers, and glacier geometry explanations are added.
Supplementary figures S2–S18 now support this analysis.
* * *
**6. Figure quality**

**Reviewer Comment:** Clarify representation, revise palettes, unify axes.

**Final Response:**
All the Figures including 3 and 4 have been updated.
Clarifications added to captions:

- Glaciers shown as polygons from RGI

- Axis ranges unified

- Colorblind-safe palettes applied

- Smoothing method specified
* * *
**7. Precipitation factor fp calibration**

**Reviewer Comment:** Explain fp and include parameter table.

**Final Response:**
We clarify that fp = 1.6 is a global OGGM value and not calibrated in this study. We have added comprehensive precipitation validation analysis against high-resolution modeling data (**Supplementary Table S1 and Figure S1**).

Explanation added in **Section 2.3.2**.

A complete parameter table is included in **Supplementary Table S2**.
* * *
**MINOR COMMENTS**
* * *
**8. Abstract clarification**

**Final Response:**
Clarified in **Abstract** (lines 17–18) that values represent ensemble mean ± 1 SD across 10 GCMs for SSP126 to SSP585.
* * *
**9. Fig. 1 MT definition and symbol clarity**

**Final Response:**
"Marine-terminating" is defined in **Figure 1 caption**.
Instead of Triangles, we used dots with reduced in size and transparency increased.
* * *
**10. Fig. 1c wording**

**Final Response:**
Changed to "Number of glaciers".
* * *
**11. Delta method explanation (L125)**

**Final Response:**
Expanded description added to **Section 2.2.1** (approx. lines 148–162).
* * *
**12. Clarify statistical test choices (L234)**

**Final Response:**
Clarified in **Section 2.4** why different tests are applied based on data distribution (lines 344–345).
* * *
**13. Figure 3 smoothing and shared y-axis**

**Final Response:**
Caption updated to state LOESS smoothing applied to mean and confidence intervals.
Shared y-axis implemented.
Changes at **Figure 3 caption**.
* * *
**14. Section 3.2 temporal and spatial comparison**

**Final Response:**
Expanded comparisons in **Section 3** and 4.
* * *
**15. Reference format consistency**

**Final Response:**
All references standardized to journal guidelines.
* * *
**16. Zenodo data level**

**Final Response:**
Clarified in **Data Availability** section (end of manuscript) that data include glacier-ID-level outputs and subregional aggregates.
* * *
**REVIEWER 2**
* * *
**MAJOR COMMENTS**
* * *
**1. Hydrological catchments vs glacier outlines**

**Final Response:**
We retain the glacier-centric fixed-gauge approach due to OGGM design constraints. Explanation added in **Section 2.3.4** (approx. lines 278-319).
* * *
**2. Regional variability and deeper analysis**

**Final Response:**
Expanded regional analysis added in **Sections 3 and 4**.

**3. Figure formatting and consistency**

**Final Response:**
All major figures reformatted for consistency.
Updates applied to **Figures 1–8**, with captions revised.
* * *
**SPECIFIC COMMENTS**
* * *
**L57 "significant mass loss process"**

**Response:**
Quantification added in **Section 1** (approx. lines 56–61).
* * *
**L70 cold-based regime explanation**

**Response:**
Brief explanation added in **Section 1** (approx. lines 73–77).
* * *
**L74ff add key dataset/method details**

**Response:**
Paragraph expanded in **Section 1** (approx. lines 82–85).
* * *
**L92 FIIC subdivision**

**Response:**
Clarified in **Section 2.1** (approx. lines 100–106).
Original RGI outline added to **Figure 1b**.
* * *
**L94 active vs inactive calving basins**

**Response:**
Definition clarified in **Section 2.1** (approx. lines 104–106).
* * *
**L105 precipitation correction**

**Response:**
Clarified in **Section 2.3.2** (approx. lines 239–254).

**L113ff GCM selection justification**

**Response:**
Explanation added in **Section 2.2.1** (approx. lines 125–145).
* * *
**L125 bias correction and interpolation**

**Response:**
Detailed clarification added in **Section 2.2.1** (approx. lines 153–162).
* * *
**L147 resolution suitable for glacier size**

**Response:**
OGGM grid resolution formula added in **Section 2.3.1** (approx. lines 200–206).
* * *
**L161ff lapse rate specification**

**Response:**
Moved climate description to **Section 2.2.1** and clarified lapse rate (constant −6.5 K km$^{-1}$) at approx. **lines 160–162**.
* * *
**L171 precipitation correction confusion**

**Response:**
Sequence clarified in revised **Section 2.2.1–2.3.5**.
* * *
**L197ff runoff terminology**

**Response:**
Clarified fixed-gauge definition in **Section 2.3.4** (approx. lines 278–316).
* * *
**L204ff peak water definition consistency**

**Response:**
Clarified glacier peak water definition in **Section 2.3.4**.
* * *
**L222 subsection title**

**Response:**
Changed to "Statistical Analysis" in **Section 2.4**.
* * *
**Fig. 3 FIIC area stability**

**Response:**
Explanation added in **Section 4.1 and 4.2**.
* * *
**L253 add Central-West volume loss**

**Response:**
Added in **Section 3.1**. (line 373)
* * *
**L256 ANOVA numbers removed**

**Response:**
Defined $p < 0.05$ in **Section 2.4**.
Removed detailed ANOVA outputs from **Section 3**.
* * *
**L290ff remove redundant sentence**

**Response:**
Sentence removed in **Section 3.1**.
* * *
**L303 runoff dominance statement**

**Response:**
Added summary paragraph in **Section 3.2** (approx. lines 431-432 & 483-489).
* * *
**L312ff predictable off-glacier increase**

**Response:**
Clarified rationale in **Section 4.2** (approx. lines 717-721).
* * *
**Fig. 7 off-glacier rain missing**

**Response:**
Updated with decimal precision. Added in **Figure 7**.
* * *
**Fig. 8 non-monotonic peak water in SE**

**Response:**
Brief explanation added in **Section 3.3** (approx. lines 531-537).
* * *
**L345 domain consistency**

**Response:**
Clarified in **Section 4.1** (approx. lines 551-553).
* * *
**L357ff NE and FIIC resilience**

**Response:**
Added detailed but concise explanation at different locations in **Section 4**.
* * *
**L376 calving-dominated wording**

**Response:**
Updated in **Section 3.2** (approx. line 490-491).
* * *
**L384 MT glacier ratios**

**Response:**
Statistics added in **Section 3.2** (approx. lines 447–459).
Time-evolving MT number and area percentages added in Figure 6b.
* * *
**L385 ocean forcing reference removed**

**Response:**
Corrected in **Section 3.2** (approx. line 477-482).
* * *
**L394 regional runoff contribution**

**Response:**
Expanded in **Section 3.2** and 4.2 (approx. lines 490-498 & 664-711).
* * *
**L403 melt season description**

**Response:**
Rephrased in **Section 4.2** (line approx. 721-723).

**L413 equilibrium statement**

**Response:**
Clarified in **Section 4.2** (approx. lines 736-742).
* * *
**L422 ocean feedback removed**

**Response:**
Corrected in **Section 4.2** (approx. line 747-749).
* * *
**L459 regional impacts**

**Response:**
Expanded in **Section 4.3** (approx. lines 780–794).
* * *
**Section 4.4 restructuring**

**Response:**
Restructured exactly as suggested.
Revised **Section 4.4** now lines approx. **805–897**.
* * *
**L537 add methods summary at start of conclusion**

**Response:**
Two-sentence summary added to **Conclusion** (approx. lines 899–904).
* * *
**L540f catchment remark**

**Response:**
Clarified glacier-centric scope in **Conclusion** (lines 909-914).
* * *
**L544–545 equilibrium vs exhaustion**

**Response:**
Clarified in **Conclusion** (approx. lines 929-935).
* * *
**L551ff expand emission-scenario differences**

**Response:**
Added to **Conclusion** (approx. lines 905-945).
* * *
**TECHNICAL COMMENTS**

All technical corrections implemented exactly as suggested:

- consistent decimals (line 33)

- smaller MT symbols (revised **Figure 1**)

- remove duplicate greys (updated **Figure 1**)

- consistent capitalization of FIIC (**revised**)

- Table 1 font uniform (revised **Table 1**)

- SIA usage standardized (**revised**)

- grammar corrections (**revised multiple locations as suggested**)

- rephrased lines (rephrased **multiple locations as suggested**)

- consistent units (**mm/yr, %/yr**) across text and figures

- Figure 3 and 4 colormaps replaced with sequential uniform colormap

- legend cleanup and regional label alignment for Figures 3–8 (revised all)

- axis consistency (y=0 start where relevant)

- explanatory note on $mm/yr^2$ added (**Section 3.1 line 403-405**)